# Age-associated differences in mucosal and systemic host responses to SARS-CoV-2 infection

Jillian H. Hurst [1,2,16], Aditya A. Mohan[3,16], Trisha Dalapati [4], Ian A. George[5], Jhoanna N. Aquino[1], Debra J. Lugo[1], Trevor S. Pfeiffer[1], Javier Rodriguez[6], Alexandre T. Rotta [7], Nicholas A. Turner [8], Thomas W. Burke [8,9], Micah T. McClain [8,9,10], Ricardo Henao [11,12], C. Todd DeMarco[13], Raul Louzao[13], Thomas N. Denny[13], Kyle M. Walsh [2,14], Zhaohui Xu [15], Asuncion Mejias [15], Octavio Ramilo [15], Christopher W. Woods [8,9,10,13] & Matthew S. Kelly [1] ✉

Age is among the strongest risk factors for severe outcomes from SARS-CoV-2 infection. Here we describe upper respiratory tract (URT) and peripheral blood transcriptomes of 202 participants (age range of 1 week to 83 years), including 137 non-hospitalized individuals with mild SARS-CoV-2 infection and 65 healthy individuals. Among healthy children and adolescents, younger age is associated with higher URT expression of innate and adaptive immune pathways. SARS-CoV-2 infection induces broad upregulation of URT innate and adaptive immune responses among children and adolescents. Peripheral blood responses among SARS-CoV-2-infected children and adolescents are dominated by interferon pathways, while upregulation of myeloid activation, inflammatory, and coagulation pathways is observed only in adults. Among SARS-CoV-2-infected individuals, fever is associated with blunted URT immune responses and more pronounced systemic immune activation. These findings demonstrate that immune responses to SARS-CoV-2 differ across the lifespan, from distinct signatures in childhood and adolescence to age-associated alterations in adults.

Age is one of the most important risk factors for severe illness and death from severe acute respiratory syndrome coronavirus 2 (SARS-CoV-2) infection. Throughout the pandemic, the majority of hospitalizations for severe coronavirus disease 2019 (COVID-19) have been among the elderly, with individuals 65 years of age or older accounting for approximately 75% of all COVID-19 deaths in the United States as of September 2023[1,2]. However, the risk of severe COVID-19 increases with age even among young adults; compared to adults 18 to 29 years of age, adults 30 to 39 years of age are approximately twice as likely to be hospitalized for COVID-19 and have four times the risk of death[3]. By comparison, children and adolescents are at relatively low risk of severe COVID-19. Between

March 2020 and September 2023, the cumulative incidence of COVID-19 hospitalization among children less than 18 years of age was 209 per 100,000 individuals, while the hospitalization rate among adults 18 years of age or older was 1797 per 100,000 individuals during this same time period[4]. Older age is also associated with higher symptom severity and a longer duration of illness among children, adolescents, and adults with mild COVID-19 not requiring hospitalization[5–7]. These differences in COVID-19 severity by age have persisted despite the introduction of vaccines and across successive waves of infections caused by distinct SARS-CoV-2 variants[8–10].

Despite the established relationship between age and COVID-19 severity, an understanding of the biological or immunological factors

---

that contribute to this association has remained elusive. Proposed hypotheses for the low risk of severe COVID-19 among children and adolescents include a lower prevalence of comorbidities[11], more recent exposure to seasonal coronaviruses[12], and age-associated differences in immunity[13,14]. Regarding the last hypothesis, several prior studies identified age-associated differences in the host response to SARS-CoV-2 infection[15–20]. However, these studies included small numbers of children, primarily focused on participants hospitalized for severe COVID-19 despite the vast majority of pediatric infections being mild, included individuals who had symptoms for weeks prior to sampling, and analyzed all pediatric age groups together, not accounting for the substantial immune maturation that occurs during infancy, childhood, and adolescence[21,22]. Additionally, relatively little is known regarding the extent to which immune responses to SARS-CoV-2 differ based on age in samples other than peripheral blood. While a few studies investigated host responses to SARS-CoV-2 within the nasal mucosa[16,19,20], these studies were similarly limited by their small sample sizes and combined analyses of pediatric age groups that included patients with varied COVID-19 severity. Thus, there remains a significant gap in our understanding of the extent to which immune responses to early SARS-CoV-2 infection vary across the full age spectrum[23]. Moreover, additional studies are needed to investigate age-associated differences in mucosal responses to SARS-CoV-2 and to understand how host responses within the upper respiratory tract (URT) influence systemic immune responses.

In this study, we investigate the URT and peripheral blood transcriptional responses of 202 participants (age range of 1 week to 83 years), including 137 non-hospitalized individuals with mild SARS-CoV-2 infection and 65 healthy individuals. SARS-CoV-2-infected participants were enrolled early in the COVID-19 pandemic, prior to the widespread circulation of major variants of concern and the routine availability of COVID-19 vaccines; thus, this analysis focuses on immune responses to SARS-CoV-2 among a population that was naïve to the virus. Moreover, we describe distinct gene expression profiles among healthy children and adolescents by age. We further provide a detailed characterization of the URT and systemic host responses to acute SARS-CoV-2 infection among multiple pediatric age groups and of systemic responses in adults. Within peripheral blood, we observe upregulation of myeloid activation, inflammatory, and coagulation pathways exclusively in SARS-CoV-2-infected adults. These data provide insights into the relationship between URT and systemic host responses to SARS-CoV-2 infection and how responses in these compartments may contribute to age-associated differences in the presentation of COVID-19, and suggest that age-related differences in host response may influence the impacts of other emerging respiratory pathogens.

## Results
### Characteristics of the study population
The samples and data included in this analysis were collected from participants in two studies conducted within the Duke University Health System (DUHS): the Biorepository of RespirAtory Virus-Exposed Kids (BRAVE Kids) study, which recruited SARS-CoV-2-exposed children and adolescents less than 21 years of age, and Molecular and Epidemiological Study of Suspected Infection (MESSI), which recruited SARS-CoV-2-exposed adults 21 years of age of older (Table 1, Supplementary Table 1). Recruitment in both studies included non-hospitalized participants who presented for SARS-CoV-2 testing within the health system and/or who had known close contact with an individual with confirmed SARS-CoV-2 infection (typically a household member). Participants in both studies were identified through review of SARS-CoV-2 testing conducted in the DUHS, and the study teams additionally approached close contacts of index cases for study participation. All participants included in this analysis were recruited between April 1, 2020, and December 31, 2020[7]. None of the

participants had a known SARS-CoV-2 infection prior to the current illness, nor had participants received a COVID-19 vaccine at the time of enrollment.

We analyzed bulk RNA sequencing data from nasopharyngeal swab ($n = 132$) and peripheral blood ($n = 155$) samples collected from 160 non-hospitalized children and adolescents (<21 years of age) and peripheral blood samples from 42 non-hospitalized adults. The median [interquartile range (IQR)] age of the pediatric participants was 11.5 (6.6, 15.7) years, 87 (54%) participants were female (Table 1). These pediatric participants were categorized into the following age categories: 0–5 years ($n = 39$; hereafter referred to as young children), 6–13 years ($n = 65$; school-age children), and 14–20 years ($n = 56$; adolescents). Of these 160 pediatric participants, 111 (69%) were classified as SARS-CoV-2-infected based on PCR testing; 49 (31%) participants were determined to be uninfected (hereafter referred to as healthy). Samples from pediatric participants were collected a median (IQR) of 4 (3, 6) days after symptom onset among symptomatic SARS-CoV-2-infected participants ($n = 63$) and 1 (0, 5) day after SARS-CoV-2 diagnosis among asymptomatic participants ($n = 48$). The median (IQR) age of the adult participants was 46.2 (40.1, 53.8) years and 21 (50%) were female (Table 1). Adult participants were similarly classified as SARS-CoV-2-infected ($n = 26$, 62%) or healthy ($n = 16$, 38%) based on PCR testing. Samples were collected a median (IQR) of 3 (3, 5) days after symptom onset among symptomatic SARS-CoV-2-infected participants ($n = 24$) and 0 (0, 0) days after SARS-CoV-2 diagnosis among asymptomatic participants ($n = 2$). The timing of sample collection was similar across age groups relative to symptom onset among symptomatic SARS-CoV-2-infected participants (Kruskal–Wallis test, $p = 0.15$) and relative to SARS-CoV-2 diagnosis among asymptomatic infected participants (Kruskal–Wallis test, $p = 0.50$).

The prevalence of most symptoms differed by age among SARS-CoV-2-infected participants (Table 1). Compared to young children and school-age children, adolescents, and adults more frequently reported cough (58% vs. 24%; Chi-square test, $p < 0.0001$), headache (52% vs. 15%; Chi-square test, $p < 0.0001$), myalgias (43% vs. 13%, Chi-square test, $p = 0.0001$), nasal congestion (38% vs. 7%, Chi-square test, $p < 0.0001$), rhinorrhea (35% vs. 7%, Chi-square test, $p < 0.0001$), loss of smell (45% vs. 6%; Fisher's exact test, $p < 0.0001$), and loss of taste (38% vs. 4%; Fisher's exact test, $p < 0.0001$). Within each age group, there were no differences in the age, sex, or prevalence of obesity among SARS-CoV-2-infected and healthy individuals (Supplementary Table 1); however, the prevalence of other comorbidities among infected adults was higher than among uninfected adults (64% vs. 19%; Fisher's exact test, $p = 0.009$). Nasopharyngeal SARS-CoV-2 viral loads were also similar among SARS-CoV-2-infected participants by age group (young children: $n = 26$, school-age children: $n = 31$, adolescents: $n = 33$, adults: $n = 25$; Kruskal–Wallis test, and $p = 0.37$) and symptom presence (symptomatic participants: $n = 72$, asymptomatic participants: $n = 43$; Wilcoxon rank-sum test, $p = 0.37$). A subset of nasopharyngeal samples from SARS-CoV-2-infected participants ($n = 28$) underwent genomic sequencing to identify the infecting lineage, with only ancestral strains being identified in these individuals (Supplementary Table 2).

### Mucosal and systemic transcriptional profiles of healthy individuals differ based on age
We first sought to describe age-associated differences in gene expression within the URT and peripheral blood of healthy young children ($n = 10$), school-age children ($n = 22$), adolescents ($n = 17$), and adults ($n = 16$, peripheral blood samples only). Because immune cell composition is known to vary with age[24], we estimated the proportions of immune cell populations within the URT and peripheral blood samples using the CIBERSORT deconvolution algorithm and the LM22 signature matrix[25]. We observed no significant differences in the imputed proportions of immune cells within the URT of healthy children and adolescents by age (Fig. 1a). Within the peripheral blood of

**Table 1 | Characteristics of the study population**

| | Young children (0–5 yr, n = 39) | School-age children (6–13 yr, n = 65) | Adolescents (14–20 yr, n = 56) | Adults (≥21 yr, n = 42) | p |
|---|---|---|---|---|---|
| **Participant characteristics** | | | | | |
| Median [IQR] age, years | 3.0 [1.6, 4.9] | 10.1 [8.4, 12.1] | 17.1 [15.6, 18.5] | 46.2 [40.1, 53.8] | <0.0001 |
| Female sex | 25 (64%) | 32 (49%) | 30 (54%) | 21 (50%) | 0.48 |
| Obesity[a] | 7 (27%) | 18 (28%) | 18 (32%) | 6 (20%) | 0.69 |
| Other comorbidities | 3 (8%) | 14 (22%) | 10 (18%) | 17 (45%) | 0.001 |
| **SARS-CoV-2 infection status** | | | | | 0.66 |
| SARS-CoV-2-infected | 29 (74%) | 43 (66%) | 39 (70%) | 26 (62%) | |
| SARS-CoV-2-uninfected (healthy) | 10 (26%) | 22 (34%) | 17 (30%) | 16 (38%) | |
| **SARS-CoV-2 illness characteristics** | | | | | |
| Asymptomatic | 14 (48%) | 20 (47%) | 14 (36%) | 2 (8%) | 0.002 |
| Fever | 13 (45%) | 16 (37%) | 16 (41%) | 14 (54%) | 0.59 |
| Cough | 7 (24%) | 10 (23%) | 19 (49%) | 19 (73%) | <0.0001 |
| Headache | 3 (10%) | 8 (19%) | 19 (49%) | 15 (58%) | <0.0001 |
| Abdominal pain | 2 (7%) | 4 (9%) | 4 (10%) | 8 (31%) | 0.06 |
| Myalgias | 0 (0%) | 9 (21%) | 13 (33%) | 15 (58%) | <0.0001 |
| Nasal congestion | 2 (7%) | 3 (7%) | 8 (21%) | 17 (65%) | <0.0001 |
| Rhinorrhea | 4 (14%) | 1 (2%) | 8 (21%) | 14 (54%) | <0.0001 |
| Loss of smell | 0 (0%) | 4 (9%) | 10 (26%) | 19 (73%) | <0.0001 |
| Loss taste | 0 (0%) | 3 (7%) | 10 (26%) | 15 (58%) | <0.0001 |
| Median [IQR] nasopharyngeal viral load, $\log_{10}$ copies/mL | 5.8 [4.0, 7.1] | 5.1 [3.5, 7.0] | 5.1 [3.7, 6.8] | 5.7 [4.6, 6.8] | 0.37 |

*IQR* interquartile range, *mL* milliliter.
[a]Children < 2 years of age were excluded from these analyses; obesity data were missing from 11 adults.

children, adolescents, and adults, we found that the proportion of transcripts associated with B cells (beta regression, $p_{adj} < 0.0001$), CD8 + T cells ($p_{adj} = 0.002$), and plasma cells ($p_{adj} = 0.005$) decreased with increasing age, while the proportions of transcripts associated with monocytes and macrophages ($p_{adj} < 0.0001$) and neutrophils ($p_{adj} = 0.0009$) increased (Fig. 1b). Because of the substantial differences in peripheral blood immune cell populations by age, we adjusted all gene expression analyses of this sample type for immune cell composition. This approach enabled us to identify genes that are differentially expressed in peripheral blood independent of age-associated differences in immune cell proportions and that more accurately reflect activation of suppression of gene expression within these cell populations. Within the URT of healthy participants, we identified 5 genes that were differentially expressed between young children and school-age children and 2 genes that were differentially expressed between school-age children and adolescents (Fig. S1a). In contrast, we identified 36 genes that were differentially expressed in young children compared to adolescents. In peripheral blood, we identified only six genes that were differentially expressed across age groups (Fig. S1b), including the metallopeptidase *ADAMTS2*, which was downregulated in young children compared to adolescents, and the transcriptional co-activator *TOX2*, which was upregulated in both school-age children and adolescents relative to adults.

To compare expression of biological pathways by age, we used the gene sets defined by the NanoString nCounter® Host Response Panel to identify groups of co-expressed genes that share a similar function (hereafter referred to as modules). We used the gene sets from this panel because they have previously been used to analyze host responses to SARS-CoV-2 and include a relatively small number of genes, enhancing the interpretability of the results[26,27]. We then used fast gene set enrichment analysis (FGSEA) to evaluate for differential expression of these modules across age groups[28,29]. Among children and adolescents, younger age groups exhibited higher expression of multiple innate immune [e.g., interferon signaling, phagocytosis, myeloid cell activation and inflammation, and natural killer (NK) cell activity] and adaptive immune (e.g., B cell receptor and T cell receptor

signaling, and lymphocyte trafficking) modules within the URT (Fig. 1c, Supplementary Data 1). Peripheral blood expression of many of these same modules was similar across pediatric age groups; however, we observed lower expression of genes associated with myeloid cell activation with decreasing age among children and adolescents (Fig. 1c, Supplementary Data 2). Compared to adults, pediatric age groups tended to have lower peripheral blood expression of interferon signaling pathways and genes associated with coagulation and myeloid cell and inflammasome activation (Fig. 1d, Supplementary Data 2).

## Distinct mucosal and systemic immune responses characterize early SARS-CoV-2 infection

To identify URT and peripheral blood transcriptional responses associated with SARS-CoV-2 infection, we initially compared gene expression profiles in SARS-CoV-2-infected versus healthy participants, combining all age groups. Using CIBERSORT to impute immune cell proportions, we found that SARS-CoV-2-infected participants had a lower proportion of dendritic cells (beta regression, $p_{adj} = 0.02$) and a higher proportion of monocytes and macrophages ($p_{adj} = 0.03$) within the URT than healthy participants (Fig. S2a), and a higher proportion of plasma cells ($p_{adj} = 0.03$) in peripheral blood (Fig. S2b). We next compared URT gene expression in infected versus healthy participants and found that SARS-CoV-2 infection was accompanied by a robust transcriptional response within the URT characterized by upregulation of gene expression for immune cell trafficking (e.g., *CCL2, CCL3, CCL4, CXCL10, and CXCL11*), interferon signaling (e.g., *IFI6, IFIT1, IFIT2, IFIT3, ISG15, NRIR, and SIGLEC1*), macrophage activity (e.g., *ACOD1, CD163, GZMB, IL6, MS4A4A, MSR1, and OLR1*), antiviral responses (e.g., *IL-27, OASL*), and autophagy (e.g., *TMEM150B*) (Fig. 2a). Gene set enrichment analyses similarly demonstrated upregulation of a broad array of innate and adaptive immune modules within the URT of SARS-CoV-2-infected participants, including multiple modules for interferon signaling (Fig. 2b, Supplementary Data 3). Compared to healthy participants, the peripheral blood gene expression of SARS-CoV-2-infected participants was characterized by upregulation of interferon-inducible genes (e.g., *ISG15, IFI27, IFI44, IFI44L, RSAD2, and SIGLEC1*) and genes

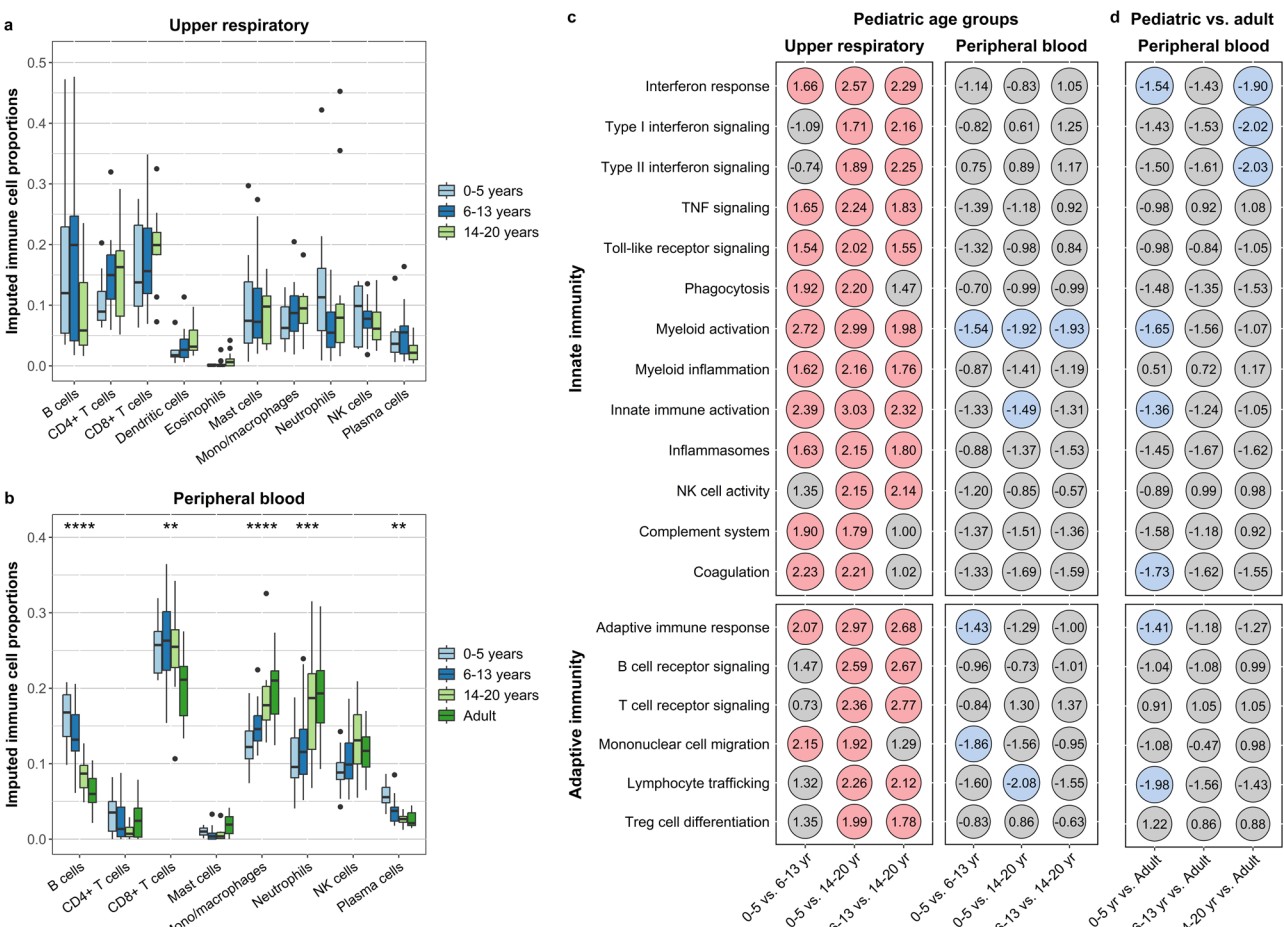

**Fig. 1 | Transcriptional profiles within the upper respiratory tract and peripheral blood of healthy children, adolescents, and adults.** Bulk RNA sequencing was used to compare the transcriptional profiles of healthy young children (0–5 years), school-age children (6–13 years), adolescents (14–20 years), and adults (≥21 years, peripheral blood samples only). **a, b** Box and whisker plots depict proportions of transcripts attributed to different immune cell populations in the upper respiratory tract and peripheral blood of healthy children and adolescents, with cell proportions imputed using CIBERSORT. Note that the proportions of cell types predicted by CIBERSORT do not reflect their absolute proportions within a given sample type. Lines splitting the boxes correspond to median values while box edges represent the 25th and 75th percentiles with outliers shown as single points. Proportions of immune cell populations were compared by age (modeled as a continuous variable) using beta regression, with all analyses corrected for multiple comparisons (*, $p_{adj} < 0.05$; **, $p_{adj} < 0.01$; ***, $p_{adj} < 0.001$; ****, $p_{adj} < 0.0001$). Only immune cell populations identified in at least 25% of samples are shown. Differential expression of gene modules in upper respiratory and peripheral blood samples from pediatric age groups (**c**) and in peripheral blood samples from children and adolescents compared to adults (**d**). Below each column, the age group listed first represents the group of interest while the age group listed second is the reference group. Numbers within each circle indicate the normalized enrichment score (NES) calculated through gene set enrichment analysis. Colored circles indicate differential expression ($p_{adj} < 0.05$) of immune modules across the comparison groups (red, upregulation in the age group of interest; blue, downregulation in the age group of interest). Gene set enrichment analyses were adjusted for sex, sequencing batch, and imputed sample immune cell proportions (peripheral blood only). Source data are provided as a Source Data file. (TNF tumor necrosis factor, NK natural killer, Treg, and regulatory T).

associated with immune cell trafficking (e.g., *CCL2, CCL8*) (Fig. 2c) Many of the same innate and adaptive immune modules upregulated within the URT of SARS-CoV-2-infected participants was also upregulated systemically in these participants, including modules associated with interferon signaling, major histocompatibility complex (MHC) class I presentation, Nod-like receptor signaling, Toll-like receptor signaling, and adaptive immune responses (Fig. 2d, Supplementary Data 3). However, modules corresponding to innate and adaptive immune cell responses and interferon response genes were generally induced to a greater degree in the URT, while the peripheral blood immune response was dominated by upregulation of interferon signaling pathways.

## Mucosal immune responses to SARS-CoV-2 infection are similar across pediatric age groups

We next sought to assess the extent to which mucosal immune responses to SARS-CoV-2 differ based on age. Using CIBERSORT, we found that the proportion of B cells decreased with increasing age

($p_{adj} = 0.003$) within the URT of SARS-CoV-2-infected participants (Fig. 3a). We then analyzed the URT transcriptional responses of SARS-CoV-2-infected participants within each pediatric age group, using healthy participants from that age group as the comparator. We observed a larger number of differentially expressed genes in infected versus healthy school-age children (377 genes) and adolescents (551 genes) relative to analyses conducted in young children (72 genes). All age groups exhibited upregulation of genes associated with chemokine signaling (e.g., *CCL2, CCL3, CCL19, CMKLR1, CXCL10,* and *CXCL11*), interferon signaling (e.g., *IFI44L, IFI6, IFIT1, IFITM1, IFITM3, ISG15,* and *SIGLEC1*), and macrophage activity (e.g., *ACOD1, CALHM6, CD163,* and *OASL*) (Fig. S3). Gene set enrichment analyses demonstrated that SARS-CoV-2 infection was associated with the upregulation of interferon responses, myeloid cell activation and inflammation, inflammasome signaling, and B and T cell receptor signaling among all pediatric age groups (Fig. 3c, Supplementary Data 4). Finally, to directly compare URT responses to SARS-CoV-2 across pediatric age groups, we used single-sample gene set enrichment analysis (ssGSEA) to generate

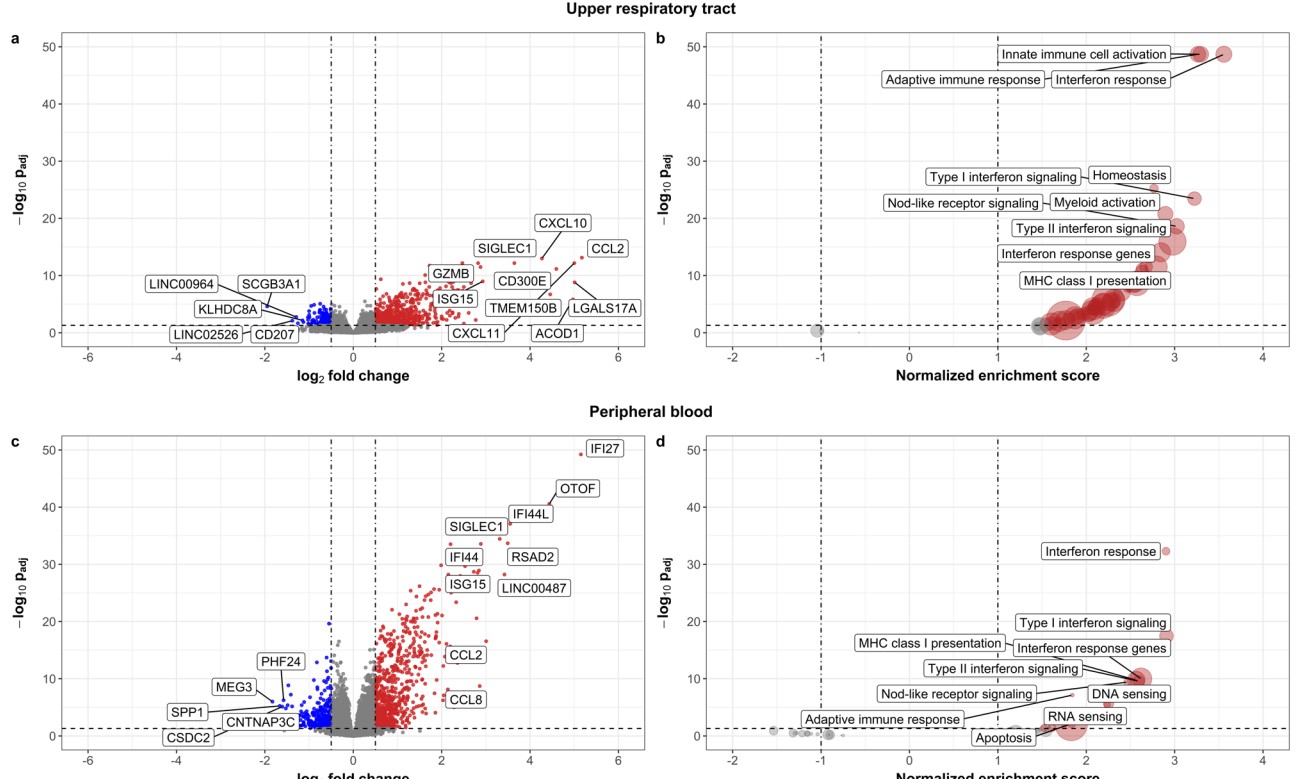

**Fig. 2 | Differential host gene expression in upper respiratory and peripheral blood samples associated with SARS-CoV-2 infection among children, adolescents, and adults.** Bulk RNA sequencing was used to compare the upper respiratory and peripheral blood transcriptional profiles of young children (0–5 years), school-age children (6–13 years), adolescents (14–20 years), and adults (≥21 years, peripheral blood only) by SARS-CoV-2 infection status. **a** Volcano plot depicting differential expression of genes in the upper respiratory tracts of SARS-CoV-2-infected compared to healthy children and adolescents ($p_{adj} < 0.05$). The ten most differentially upregulated and 5 most downregulated genes in SARS-CoV-2-infected participants based on log2-fold change are labeled. **b** Volcano plot depicting differential expression of gene modules in upper respiratory samples by SARS-CoV-2 infection status. The ten modules with the highest normalized enrichment scores

(NES; upregulated in SARS-CoV-2 infection) are labeled; no modules were downregulated in SARS-CoV-2-infected participants. **c** Volcano plot depicting differential expression of genes in the peripheral blood of SARS-CoV-2-infected compared to healthy children, adolescents, and adults. The 10 most differentially upregulated and 5 most downregulated genes in SARS-CoV-2-infected participants based on log2-fold change are labeled. **d** Volcano plot depicting differential expression of gene modules in peripheral blood by SARS-CoV-2 infection status. The 10 modules with the highest NES (upregulated in SARS-CoV-2-infected participants) are labeled; no modules were downregulated in SARS-CoV-2-infected participants. All analyses shown were adjusted for sex, sequencing batch, age group, and imputed sample immune cell proportions (peripheral blood only). Source data are provided as a source data file. (MHC major histocompatibility complex).

enrichment scores corresponding to the levels of expression of immune modules within each sample. We then fit linear regression models with module enrichment scores as the dependent variables and included an interaction term between SARS-CoV-2 infection status and age group in these models, enabling direct comparison of changes in immune module expression associated with SARS-CoV-2 infection by age group. Using this approach, we did not observe any significant differences in the degree to which these modules were upregulated within the URT in association with SARS-CoV-2 infection across pediatric age groups (Fig. 3c).

### Systemic immune responses to SARS-CoV-2 infection differ among children, adolescents, and adults

We next performed similar analyses among pediatric age groups and adults to evaluate for differences in systemic immune responses to SARS-CoV-2 based on age. Using CIBERSORT to infer immune cell proportions, we identified differences in multiple immune cell populations in the peripheral blood of SARS-CoV-2-infected participants by age (Fig. 3b). Specifically, we found that the proportions of transcripts associated with several innate cell populations increased with age among infected participants [monocytes and macrophages ($p_{adj} < 0.0001$), neutrophils ($p_{adj} < 0.0001$), NK cells ($p_{adj} = 0.02$)], while the proportions of transcripts associated with specific adaptive immune cell populations decreased with age [B cells ($p_{adj} < 0.0001$),

CD8+ T cells ($p_{adj} = 0.0003$), plasma cells ($p_{adj} = 0.001$)]. These age-associated differences in imputed peripheral blood immune cell populations among SARS-CoV-2-infected participants were similar to those observed in healthy participants (Fig. 1b). We then compared the peripheral blood transcriptional profiles of SARS-CoV-2-infected versus healthy participants within each age group, adjusting for differences in imputed immune cell proportions. We found that the expression of interferon signaling pathways was upregulated in SARS-CoV-2-infected participants of all age groups compared to healthy controls (Fig. 3c, Supplementary Data 5, Fig. S4). Young children exhibited downregulation of transcripts associated with myeloid cell activation and lymphocyte trafficking, while school-aged children exhibited downregulation of transcripts associated with leukotriene and prostaglandin inflammation and NK cell activity (Fig. 3c, Supplementary Data 5). In contrast, only adults with SARS-CoV-2 infection demonstrated upregulation of transcripts associated with the complement system, coagulation, innate immunity (e.g., myeloid activation, phagocytosis, and toll-like receptor signaling), inflammation (e.g., inflammasomes, tumor necrosis factor (TNF) signaling), and adaptive immunity (e.g., B cell receptor signaling, mononuclear cell migration, and lymphocyte trafficking) (Fig. 3c, Supplementary Data 5).

We then used ssGSEA and linear regression to compare the systemic immune responses associated with SARS-CoV-2 infection across age groups. Compared to other pediatric age groups and adults, young

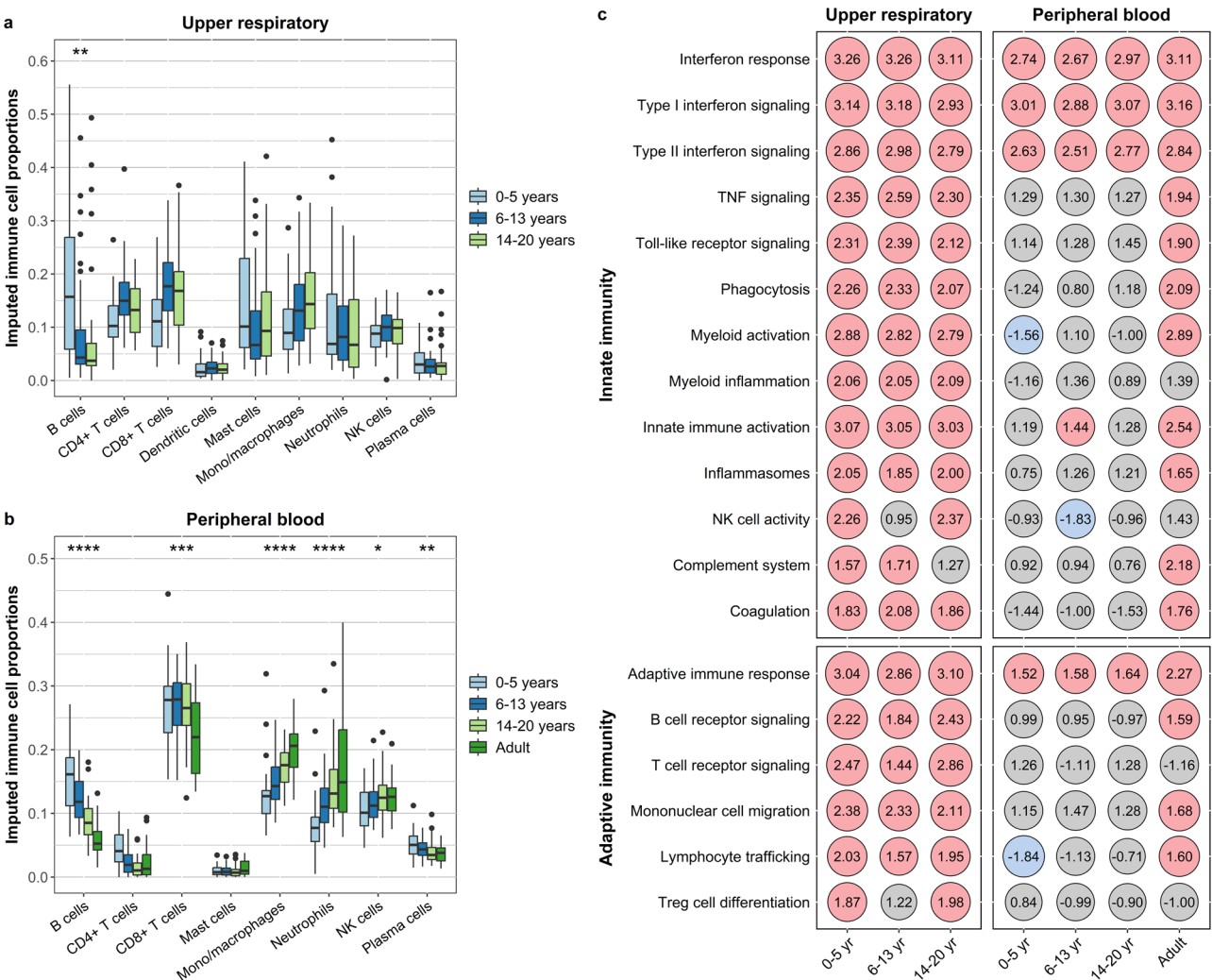

**Fig. 3 | Transcriptional profiles associated with SARS-CoV-2 infection by age group.** Bulk RNA sequencing was used to compare the upper respiratory and peripheral blood transcriptional profiles of young children (0–5 years) school-age children (6–13 years), adolescents (14–20 years), and adults (≥21 years; peripheral blood only). **a**, **b** Box and whisker plots depict proportions of transcripts attributed to different immune cell populations in the upper respiratory tract and peripheral blood of SARS-CoV-2-infected participants, with cell proportions imputed using CIBERSORT. Note that the proportions of cell types predicted by CIBERSORT do not reflect their absolute proportions within a given sample type. Lines splitting the boxes correspond to median values while box edges represent the 25th and 75th percentiles with outliers shown as single points. Proportions of immune cell populations were compared by age (modeled as a continuous variable) using beta regression, with all analyses corrected for multiple comparisons (*, $p_{adj} < 0.05$; **, $p_{adj} < 0.01$; ***, $p_{adj} < 0.001$; ****,

$p_{adj} < 0.0001$). Only immune cell populations identified in at least 25% of samples are shown. **c** Differential expression of gene modules in the upper respiratory tract and peripheral blood associated with SARS-CoV-2 infection by age group. Within each age group listed below the columns, module expression among SARS-CoV-2-infected participants was compared to that of healthy participants in that same age group. Numbers within each circle indicate the normalized enrichment score (NES) calculated through gene set enrichment analysis. Colored circles indicate differential expression ($p_{adj} < 0.05$) of gene modules by SARS-CoV-2 infection status (red, upregulated in SARS-CoV-2-infected participants; blue, downregulated in SARS-CoV-2-infected participants). Gene set enrichment analyses were adjusted for sex, sequencing batch, and imputed sample immune cell proportions (peripheral blood only). Source data are provided as a Source Data file. (TNF tumor necrosis factor; NK natural killer; Treg and regulatory T).

children had less systemic upregulation of genes corresponding to myeloid cell activation (linear regression; vs. school-age children: $p = 0.04$, vs. adolescents: $p = 0.02$, vs. adults: $p = 0.048$), phagocytosis (vs. school-age children: $p = 0.03$, vs. adolescents: p = 0.0005, vs. adults: $p = 0.07$), and lymphocyte trafficking (vs. school-age children: $p = 0.002$, vs. adolescents: $p = 0.008$, vs. adults: $p = 0.03$) associated with SARS-CoV-2 infection. Comparing adults to pediatric age groups, SARS-CoV-2 infection induced greater peripheral blood upregulation of genes involved with the complement system (vs. young children: $p = 0.01$, vs. school-age children: $p = 0.051$, vs. adolescents, $p = 0.14$), and less upregulation of genes corresponding to T cell receptor signaling (vs. young children, $p = 0.01$, vs. school-age children: $p = 0.04$, vs. adolescents, $p = 0.002$) and regulatory T cell differentiation (vs.

young children, $p = 0.04$, vs. school-age children, $p = 0.08$, vs. adolescents, $p = 0.009$). These findings demonstrate that the systemic immune responses to SARS-CoV-2 infection differ across the age spectrum, including between pediatric age groups.

## Mucosal and systemic immune responses to SARS-CoV-2 differ by illness characteristics

Because we and others have previously observed marked differences in the symptoms of SARS-CoV-2 infection across age groups[7,30–32], we evaluated associations between specific symptoms and transcriptional responses within the URT and peripheral blood of SARS-CoV-2-infected participants. Specifically, we performed age-adjusted analyses comparing the expression of specific immune modules in SARS-

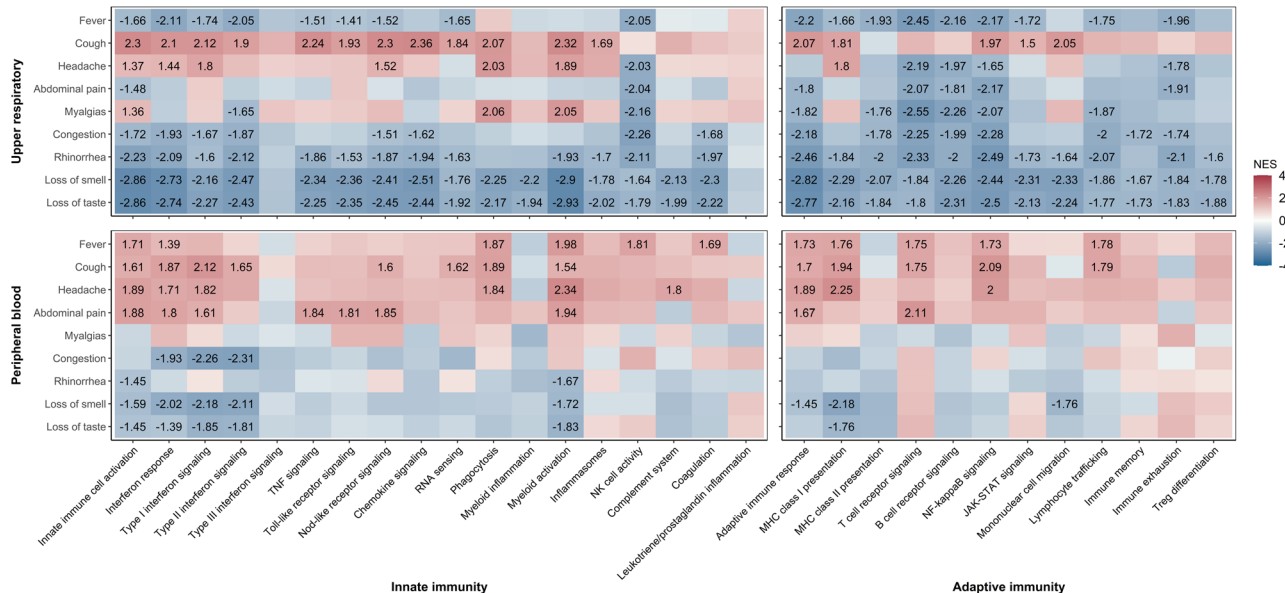

**Fig. 4 | Differential expression of immune modules in SARS-CoV-2 infection by symptom presence.** Heatmaps depict results of gene set enrichment analysis of upper respiratory (children and adolescents) and peripheral blood (children, adolescents, and adults) bulk RNA sequencing data comparing gene expression among SARS-CoV-2-infected participants who reported a symptom to infected participants who did not report that symptom. Each column corresponds to a gene module comprised of co-expressed genes that share a similar function. The numbers within blocks represent normalized enrichment scores (NES) for comparisons of gene module expression that were statistically significant ($p_{adj} < 0.05$). The color of each block indicates the direction of the change in expression (red, upregulated in the condition of interest; blue, downregulated in the condition of interest). Analyses were adjusted for age group, sex, sequencing batch, and imputed sample immune cell proportions (peripheral blood only). Source data are provided as a source data file. (TNF tumor necrosis factor; NK natural killer; Treg regulatory T).

CoV-2-infected participants who reported a symptom at any point during the illness course to infected participants who did not report that symptom (Fig. 4, Supplementary Data 6 and 7). Participants who reported fever exhibited downregulation of innate immune (e.g., interferon signaling, TNF signaling, Nod-like receptor signaling, RNA sensing, and NK cell activity) and adaptive immune pathways (e.g., MHC class I/II presentation, T and B cell receptor signaling, NF-κB signaling, JAK/STAT signaling, lymphocyte trafficking, immune memory, and immune exhaustion) in the URT, and the upregulation of many of these same pathways (e.g., interferon signaling, NK cell activity, MHC class I, T cell receptor signaling, NF-κB signaling, and lymphocyte trafficking) in peripheral blood. In contrast, participants who reported cough or headache exhibited broad upregulation of innate and adaptive immune signaling pathways in both the URT and peripheral blood, including modules corresponding to innate immune cell activation, interferon signaling, phagocytosis, myeloid cell activation, and MHC class I presentation. Participants reporting symptoms localizing to the URT, including rhinorrhea, nasal congestion, and loss of smell or taste, exhibited broad downregulation of innate and adaptive immune responses in the URT and downregulation of interferon signaling pathways in peripheral blood.

### Correlations between mucosal and systemic immune responses in acute SARS-CoV-2 infection

Given that the URT serves as the primary entry point for SARS-CoV-2, we hypothesized that a strong antiviral response within the URT would be associated with blunted systemic immune responses. We, therefore, evaluated correlations between the expression of immune modules in paired URT and peripheral blood samples collected from pediatric participants. We observed strong positive correlations between expression of both interferon signaling and RNA sensing pathways in URT and peripheral blood samples (Fig. 5, Supplementary Data 8). In contrast, higher URT expression of other innate immune modules, including those corresponding to phagocytosis and myeloid cell activation, was associated with lower expression of innate immune pathways within peripheral blood. Similarly, URT expression of adaptive immune modules, including B cell receptor signaling, NK cell activity, and immune memory pathways, was negatively correlated with peripheral blood expression of several gene modules, including those for myeloid cell activation, coagulation, and lymphocyte trafficking. Notably, we observed positive correlations between interferon signaling and adaptive immune responses in the URT and the expression of genes associated with immune memory in peripheral blood.

## Discussion

In this study, we performed bulk RNA sequencing of URT and peripheral blood samples from 202 individuals across the lifespan, evaluating host gene expression in one of the largest and most age-diverse cohorts studied to date. Importantly, the size of this cohort enabled us to compare host responses to the virus across pediatric age groups previously shown to have varied COVID-19 illness presentations. Moreover, this cohort was recruited early in the COVID-19 pandemic, providing insights into host responses to SARS-CoV-2 in a population that was largely naïve to the virus. Unlike many prior studies, our cohort only included individuals who did not require hospitalization for COVID-19 and is therefore representative of the vast majority of SARS-CoV-2 infections, particularly among children and adolescents. We demonstrate that there are distinct host responses to early SARS-CoV-2 infection across the age spectrum, including among children and adolescents. Moreover, we describe relationships between compartment-specific immune responses within the URT and peripheral blood that likely contribute to observed differences in the clinical presentation of SARS-CoV-2 infection among children, adolescents, and adults.

One of the most striking epidemiological features of the COVID-19 pandemic has been the variable disease severity across age groups. In particular, children and adolescents have been at low risk of severe disease, suggesting that important insights can be gained by studying the immune responses of pediatric age groups to SARS-CoV-2

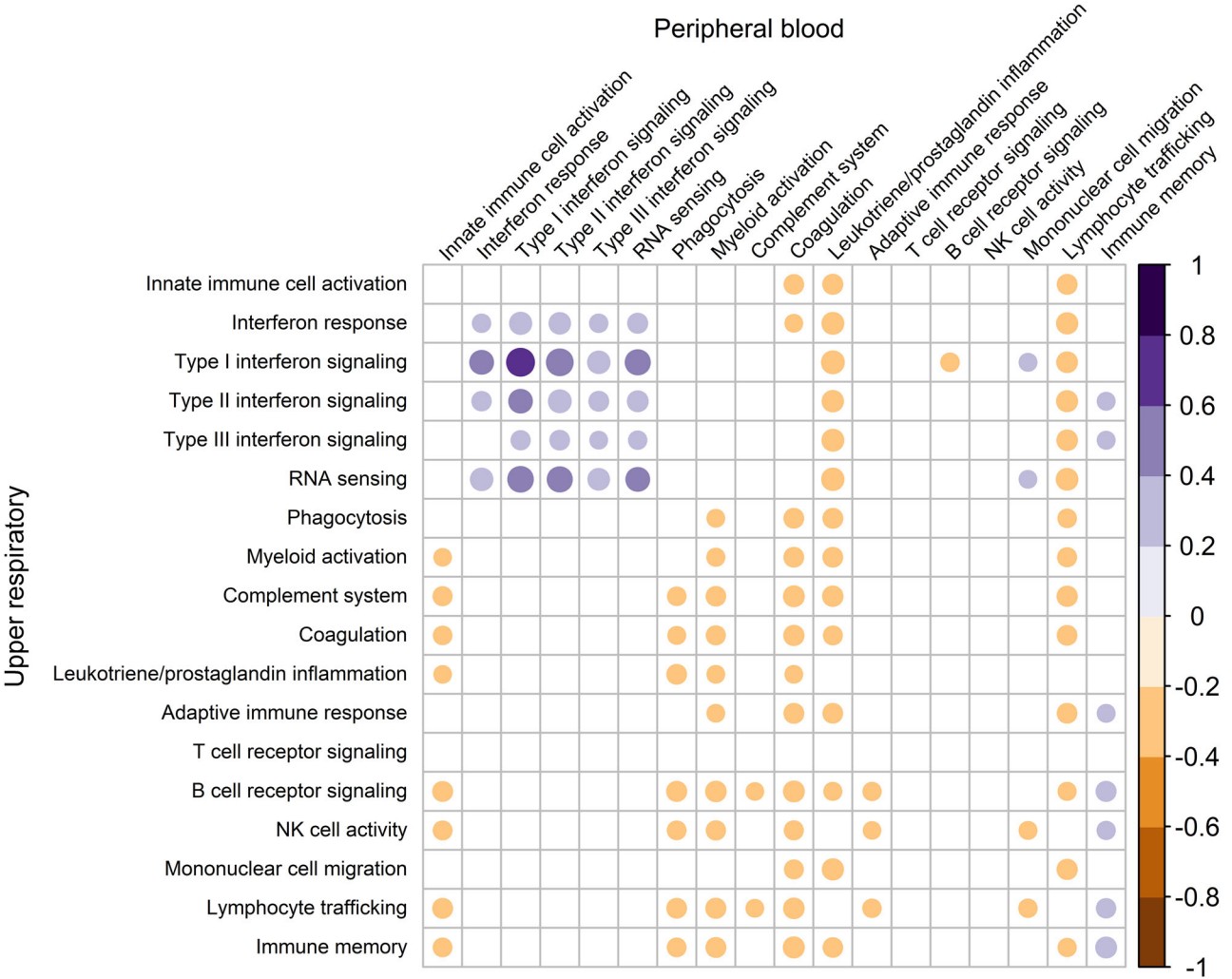

**Fig. 5 | Correlations between immune module expression in the upper respiratory tract and peripheral blood of SARS-CoV-2-infected children and adolescents.** Single-sample gene set enrichment analysis was used to calculate enrichment scores for gene modules in paired upper respiratory and peripheral blood samples from SARS-CoV-2-infected children and adolescents. Pearson's correlation coefficients were then calculated to evaluate for linear relationships between expression of these modules in the upper respiratory tract and peripheral blood of the same individual. Positive correlations are displayed in purple and negative correlations are displayed in orange. The size and color of each circle correspond to the strength of the correlation; only statistically significant correlations after adjustment for multiple comparisons ($p_{adj} < 0.05$) are shown. Source data are provided as a source data file. (NK, natural killer).

infection. Prior studies of systemic host responses to SARS-CoV-2 have noted consistent differences by age, but these studies have included relatively few children and adolescents and have not evaluated for differences between pediatric age groups. Yoshida and colleagues used single-cell RNA sequencing to analyze respiratory and peripheral blood samples from 19 children (<18 years of age) and 18 adults hospitalized for COVID-19, with nearly half of each age group requiring respiratory support. They observed that children had less robust systemic interferon responses and exhibited a more naïve and less cytotoxic T cell response to SARS-CoV-2 infection than adults[16]. Vono and colleagues observed that 16 children and adolescents (0 to 16 years of age) mounted a faster, more transient systemic immune response to SARS-CoV-2 infection than 21 adults, though children and adults exhibited similar expression of genes associated with interferon responses, immune cell activation, and inflammation[17]. Finally, Pierce and colleagues reported that 65 children, adolescents, and young adults (<24 years of age) hospitalized for COVID-19 had lower serum concentrations of inflammatory cytokines and muted peripheral blood cellular immune responses relative to 60 hospitalized adults 24 years of age or older[15]. In our cohort, which included 137 non-hospitalized participants with mild SARS-CoV-2 infection, children had less

systemic activation of several innate immune and inflammatory pathways in response to SARS-CoV-2 than adults. In particular, SARS-CoV-2 infection was associated with upregulation of genes corresponding to myeloid cell activation, complement, coagulation, and inflammatory signaling pathways in the peripheral blood of adults, while expression of these genes was unchanged among infected children and adolescents. Prior studies implicated myeloid cell dysregulation in the pathogenesis of severe COVID-19[33-35], suggesting that the observed differences in the expression of myeloid cell-associated genes between children and adults may contribute to the varied illness presentation in these age groups. Moreover, consistent with the clinical observation that children and adolescents with COVID-19 are at lower risk of thrombotic complications than adults[36], we found that even mild SARS-CoV-2 infection was associated with upregulation of coagulation genes among adults, while these pathways were unchanged in infected children and adolescents.

The URT is the primary point of entry for SARS-CoV-2 and most other respiratory viruses[37]; thus, an improved understanding of host responses to the virus within this niche could identify biological processes that influence infection susceptibility, illness severity, and systemic immune responses to the virus. To date, the majority of studies

evaluating host responses to SARS-CoV-2 infection have focused on systemic responses or responses within specific immune cell subsets[38]. Analyzing URT samples from healthy children and adolescents without SARS-CoV-2 infection, we found that younger age was associated with broad upregulation of innate and adaptive immune pathways. Only a few prior studies have investigated age-associated differences in mucosal immunity among healthy individuals. Pierce and colleagues performed bulk RNA sequencing of nasopharyngeal samples from 12 children and 27 adults with SARS-CoV-2 infection, observing higher expression of interferon signaling, inflammasome, and innate immune pathways among children relative to adults[19]. Of note, the adults in this cohort were more likely to be hospitalized than the children; thus, differences in disease severity or the timing of sampling across age groups could have contributed to the observed differences in URT immune responses[19]. Yoshida and colleagues reported that adults with SARS-CoV-2 infection had stronger induction of interferon-related genes within the URT than children; however, this cohort also included patients with varied COVID-19 disease severity[16]. Using an age-matched cohort of non-hospitalized, SARS-CoV-2-infected and uninfected children and adults, Loske and colleagues observed that children had greater upregulation of interferon responses within the URT than adults[18]. Taken together, these findings suggest that children may be primed to rapidly respond to exogenous pathogens, with the potential for earlier local control of infection and reduced illness severity.

As we and others have shown, effective mucosal immune responses to respiratory viruses can limit viral replication and reduce systemic immune activation[18,39]. Several prior studies demonstrated a relationship between mucosal immune responses and the severity of respiratory viral infections. In an influenza household transmission study, Oshansky and colleagues reported associations between age, the nasal cytokine response, and disease severity, with innate immune responses in the respiratory mucosa being the strongest predictor of clinical outcomes[40]. In a study of 37 children with rhinovirus or respiratory syncytial virus (RSV) lower respiratory infection, García and colleagues found that nasal cytokine concentrations were inversely correlated with disease severity, suggesting a protective effect of a robust immune response within the URT[41]. Similarly, Taveras and colleagues observed that children with mild RSV infection had higher levels of mucosal interferons than children with more severe disease[42]. Koch and colleagues directly compared host responses of children infected with SARS-CoV-2, influenza, or RSV, and found a high degree of similarity in gene expression within the nasal mucosa across these viral infections[20]. RSV and SARS-CoV-2 infections both elicited expression of gene modules associated with inflammatory responses, T cell activation, and IL-6 production, while influenza and SARS-CoV-2 infections elicited expression of gene modules associated with antiviral responses, immune cell recruitment, and type I interferon signaling. Notably, upregulation of only one gene module was exclusively associated with SARS-CoV-2 infection; however, this module did not correlate with illness severity or outcomes among SARS-CoV-2-infected children[20]. These findings suggest that systemic immune responses may drive differences in illness severity and outcomes of these viral infections, including across different age groups.

We and others have evaluated for associations between URT and peripheral responses. Using paired nasopharyngeal and plasma samples from 49 adults hospitalized for COVID-19, Smith and colleagues found that greater disease severity was associated with decreased interferon expression in the URT and higher plasma levels of inflammatory cytokines, suggesting that a blunted URT immune response may result both in a heightened systemic immune response and worse clinical outcomes[43]. Although the SARS-CoV-2-infected participants in our cohort had mild infections not requiring hospitalization, we observed that the presence of fever was associated with diminished innate and adaptive immune responses within the URT and increased systemic immune activation. We additionally found that SARS-CoV-2-infected participants with symptoms localizing to the URT, including rhinorrhea and loss of taste or smell, had downregulated URT, and peripheral immune responses compared to infected participants who did not have these symptoms. These findings are consistent with previous studies demonstrating that loss of taste and smell are associated with mild COVID-19[44,45].

Our study has several strengths and weaknesses. Strengths of the study include the large sample size and inclusion of individuals across a wide age spectrum. Additionally, this study focused on an outpatient population with mild SARS-CoV-2 infection that was enrolled prior to the routine availability of COVID-19 vaccines and the emergence of major SARS-CoV-2 variants, providing a unique opportunity to compare mucosal and systemic host responses to SARS-CoV-2 by age in a population largely naïve to the virus. Weaknesses of the study include a lack of SARS-CoV-2-infected participants with severe disease, the inclusion of which could have provided insights into host responses associated with poor outcomes among different age groups. Additionally, we only collected samples from participants once during early infection; prior studies suggest that mucosal and systemic immune responses to SARS-CoV-2 are dynamic during acute infection[46,47]. We relied on bulk RNA sequencing of URT and peripheral blood samples, which supported compartment-specific comparisons of gene expression but precluded us from evaluating gene expression within specific tissues or immune cell subsets. Analyses of URT gene expression were limited to children and adolescents because adult nasopharyngeal samples were collected in a medium that did not sufficiently preserve RNA. This lack of data from adult URT precluded evaluation of the relationship between mucosal and systemic immune responses to SARS-CoV-2 in this age group. Illness severity was classified based upon symptom prevalence, as testing location and medically attended visits were not reliable measures of disease severity in this cohort of individuals with mild or asymptomatic infections recruited early in the COVID-19 pandemic. Finally, this analysis focused on participants infected with SARS-CoV-2 early in the pandemic, which may limit the generalizability of our findings to infections caused by currently circulating SARS-CoV-2 variants in a population with substantial herd immunity. Conversely, this SARS-CoV-2-naïve population presents a unique opportunity to understand how individuals respond to SARS-CoV-2 infection in the absence of prior immune experience, and may also provide important insights into how age influences responses to other emerging viral pathogens.

In conclusion, we found that age was strongly associated with host responses to mild SARS-CoV-2 infection. Among children and adolescents, younger age was associated with a greater degree of URT immune activation in healthy children and with more robust URT immune activation in response to SARS-CoV-2 infection. Correspondingly, children and adolescents exhibited a relatively muted systemic immune response to SARS-CoV-2 infection compared to adults. Moreover, we identify mucosal and systemic host responses associated with specific COVID-19 symptoms and provide evidence that strong mucosal immune responses are associated with less systemic immune activation. Our findings demonstrate that host responses to SARS-CoV-2 vary substantially by age, including among children and adolescents, and contribute to observed differences in the clinical presentation of COVID-19 across the lifespan.

## Methods

### Regulatory approvals

The relevant protocols were approved by DUHS Institutional Review Board (Pro00106150, Pro00100241). Informed consent was obtained from all study participants or their legal guardians, with written approval obtained using an electronic consent document. Informed consent was obtained from all study participants, with written approval obtained using an electronic consent document. All study

protocols were conducted in accordance with the Declaration of Helsinki, applicable regulations, and local policies.

## Study design

The children and adolescents (<21 years) included in these analyses were enrolled in the Biospecimens from RespirAtory Virus-Exposed Kids (BRAVE Kids) study, a prospective cohort study of non-hospitalized individuals with confirmed SARS-CoV-2 infection or close contact with an individual with confirmed infection[7]. Participants were identified either through presentation to the health system for SARS-CoV-2 testing or through identification of close contact with PCR-confirmed SARS-CoV-2 infection. At the time of enrollment, a study team member administered a questionnaire by telephone to the participant or a caregiver to gather information on sociodemographic factors, potential sources of SARS-CoV-2 exposure, and relevant past medical history. Questionnaires were also administered to assess the presence and duration of specific symptoms at enrollment and at 7, 14, and 28 days after enrollment or until participants reported resolution of all symptoms. Nasopharyngeal samples were collected with nylon flocked swabs (Copan Italia, Brescia, Italy), placed into RNAProtect (Qiagen, Hilden, Germany), and tested for SARS-CoV-2 by quantitative PCR[7]. Whole blood samples were collected into PAXgene blood RNA tubes (Qiagen).

The adults (≥21 years) included in these analyses participated in Molecular and Epidemiological Study of Suspected Infection (MESSI), a prospective cohort study of individuals with confirmed SARS-CoV-2 infection or close contact with an individual with confirmed infection. Although this study additionally enrolled individuals hospitalized for COVID-19, only non-hospitalized individuals were included in the present analyses. Nasopharyngeal samples were collected using pre-packaged kits containing nylon flocked swabs and viral transport medium (VTM; Dasky Life Science, Ningbo, China). Whole blood samples were collected into PAXgene blood RNA tubes (Qiagen). Detailed data on symptoms, exposures, and medical history were similarly collected from MESSI study participants using serial questionnaires.

While the children and adults in this analysis were enrolled into separate study protocols, the two study teams coordinated study activities to the extent possible, frequently conducting joint visits to households with multiple eligible participants and using the same sample collection and processing procedures. The primary difference in the procedures of these two studies was in the medium used for nasopharyngeal swab samples; samples in the BRAVE Kids study were collected into RNAProtect and samples in the MESSI study were collected into VTM, with the latter medium precluding transcriptomic analyses. Nasopharyngeal and blood samples from both studies were initially processed and stored in the Duke Human Vaccine Institute Accessioning Unit. These samples underwent RNA extraction and library preparation using the same workflow in the Duke University Sequencing and Genomic Technologies core facility. Finally, whenever possible, samples from both studies were included in sequencing batches to ensure that results could be compared across studies.

Participants from both studies were classified as SARS-CoV-2-infected if the virus was detected by PCR testing of nasopharyngeal swabs collected for clinical or research purposes. All samples from SARS-CoV-2-infected participants included in the analyses presented herein were collected within 14 days of symptom onset or SARS-CoV-2 diagnosis, whichever came first. Additionally, the inclusion of individuals with SARS-CoV-2 infection was limited to participants who did not require hospitalization, had no prior history of COVID-19 vaccination, and who were enrolled between April 1, 2020, and December 31, 2020, prior to widespread circulation of major SARS-CoV-2 variants of concern. Participants who tested negative for SARS-CoV-2 but who reported one or more symptoms at enrollment or during study follow-up were also excluded because these symptoms could be indicative of false-negative SARS-CoV-2 PCR testing or infection by other adventitious agents.

## RNA sequencing and bioinformatic processing of sequencing reads

Total RNA was extracted from whole blood PAXgene tubes using PAXgene blood miRNA extraction kits (Qiagen). Nasopharyngeal samples from the BRAVE Kids study were extracted using RNeasy kits (Qiagen). RNA quantity and quality were assessed using a NanoDrop 2000 spectrophotometer (Thermo Fisher Scientific, Waltham, MA) and a Bioanalyzer 2100 system with RNA 6000 Nano kits (Agilent, Santa Clara, CA). Median (IQR) RNA integrity numbers for nasopharyngeal and peripheral blood samples were 61 (4.7–7.2) and 8.1 (7.5–8.4), respectively. Library preparation was performed using Universal Plus mRNA library kits (Tecan, Männedorf, Switzerland) with AnyDeplete Globin (NuGEN Technologies, Redwood City, CA). The resulting libraries were sequenced on a NovaSeq 6000 instrument (Illumina, San Diego, CA) configured for 100 base-pair paired-end sequencing. Raw sequencing reads were assessed for quality using FastQC v0.11.9 and trimmed using Trimmomatic v0.39[48,49]. Median (IQR) sequencing depth for nasopharyngeal and peripheral blood samples after quality filtering was 41.6 (33.8–55.4) and 49.0 (39.0–55.7) million reads, respectively. The resulting quality-filtered reads were aligned to the human reference genome GRCh38 and a gene count matrix was generated using STAR v2.7.10b[50]. Gene annotation was performed using the GENCODE v43 basic annotation[51]. Genes not expressed at a level of at least 1 count per million in at least 50% of samples were excluded from further analyses.

## SARS-CoV-2 genome assembly and lineage assignment

Total RNA was extracted from nasopharyngeal samples using QIAamp Viral RNA Mini kits (Qiagen), and cDNA synthesis was performed with random hexamers. The SARS-CoV-2 genome was amplified using COVIDSeq RUO kits (Illumina), as per the manufacturer's instructions. The resulting amplicons were then used to build standard libraries that were sequenced on a NextSeq 500 instrument (Illumina) configured for 75 base-pair single-end sequencing. Raw data were processed using an automated pipeline (https://github.com/wodanaz/Assembling_viruses) that performed several quality control procedures (de-duplication, read quality filtering, removal of off-target reads and barcode swaps), generated genome assemblies, and used Pangolin v3.1.20 for lineage assignment[52].

## Statistical analyses

We used Chi-squared or Fisher's exact tests (categorical variables) and Wilcoxon rank-sum or Kruskal–Wallis tests (continuous variables) to evaluate for differences in sociodemographic and clinical characteristics by age group and SARS-CoV-2 infection status. To estimate the proportions of immune cell populations in URT and peripheral blood samples, we used the CIBERSORT deconvolution algorithm and the LM22 gene signature matrix[25]. Briefly, CIBERSORT is an analytical tool that uses support vector regression to estimate the relative proportions of cell types with samples using gene expression data. Note that the proportions of cell types predicted by CIBERSORT do not reflect their absolute proportions within a given sample type. For this analysis, CIBERSORT was run in relative mode with B-mode batch correction and the number of permutations set to 500. LM22 is a validated gene signature matrix consisting of 547 genes that were previously shown to accurately differentiate 22 immune cell populations, including 11 major leukocyte types[25]. We compared the proportions of immune cell populations by age (modeled as a continuous variable) and SARS-CoV-2 infection status using beta regression[53]. We observed substantial differences in imputed immune cell populations in peripheral blood samples according to age; thus, we chose to adjust all peripheral blood gene expression analyses for imputed immune cell composition. To

reduce the dimensionality of these data, we generated principal components for peripheral blood samples using the *stats* R package and data for the 11 major leukocyte types in the LM22 signature matrix. We then adjusted peripheral blood gene expression analyses for the first five principal components generated for each sample type, accounting for 83% of the variance in the proportions of major leukocyte types across peripheral blood samples. We additionally adjusted these analyses for sequencing batch and participant sex and, when not the comparison of interest, we adjusted for or stratified by age group and SARS-CoV-2 infection status. We used DESeq2 to evaluate for differences in the expression of individual genes across groups of interest[54].

We identified modules comprised of groups of co-expressed genes that share a similar function using the gene sets defined by the NanoString nCounter® Host Response Panel (Supplementary Data 9)[29]. We then used FGSEA to evaluate for differential expression of these modules across comparisons of interest[28]. We used ssGSEA to generate module enrichment scores for each sample among SARS-CoV-2-infected children and adolescents. To directly compare immune responses to SARS-CoV-2 across age groups, we fit a linear regression model for each immune module with the enrichment score as the dependent variable and an interaction term between SARS-CoV-2 infection status and age group as an independent variable; these analyses were additionally adjusted for sequencing batch, participant sex, and imputed immune cell proportions (peripheral blood samples only). We then calculated Pearson's correlation coefficients to evaluate for linear relationships between expression of immune modules within the URT and peripheral blood of the same individual[55]. All analyses were corrected for the false discovery rate due to multiple tests using the Benjamini–Hochberg procedure. Analyses were performed using R version 4.4.2[56].

## Statistics and reproducibility

The study described herein was an epidemiologic investigation of SARS-CoV-2 exposure and infection among children and adults in central North Carolina. Individuals were identified through presentation to the health system for SARS-CoV-2 testing or through identification of close contact with PCR-confirmed SARS-CoV-2 infection. No statistical method was used to predetermine sample size, experiments were not randomized, and the investigators were not blinded. Participants who tested negative for SARS-CoV-2 but reported one or more symptoms at enrollment or during study follow-up were excluded from analysis due to the potential for false-negative SARS-CoV-2 PCR results or infection with other adventitious agents. No other data were excluded from the analyses.

## Reporting summary

Further information on research design is available in the Nature Portfolio Reporting Summary linked to this article.

## Data availability

The RNA sequencing dataset supporting the conclusions of this study is available in the Gene Expression Omnibus (accession number: GSE231409; https://www.ncbi.nlm.nih.gov/geo/query/acc.cgi?acc=GSE231409). The sequencing dataset used for SARS-CoV-2 lineage assignment is available in the Sequence Read Archive (PRJNA1024980; https://www.ncbi.nlm.nih.gov/sra/?term=PRJNA1024980). The de-identified clinical metadata file is available at: https://github.com/mskelly7/COVID_RNASeq_Age/blob/main/Data_Files/BRAVE_RNASeq_Metadata.csv. Source data are provided with this paper.

## Code availability

The statistical files and scripts used for data analyses are publicly available (https://github.com/mskelly7/COVID_RNASeq_Age).

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

## Acknowledgements

We offer sincere gratitude to the children and families who participated in this research. We additionally thank Dr. Nicolas Devos, Dr. Devjanee Swain Lenz, Sarah Clarke, and the staff at the Sequencing and Genome Technologies core facility at Duke University for sample processing, library construction, and genome sequencing. We thank Dr. Alejandro Berrio Escobar, John Bradley, Hilmar Lapp, and Dr. Gregory Wray for analysis of SARS-CoV-2 genome sequences and variant calls. This research was supported by a Merck Investigator Studies Program Grant (MISP #60495 to MSK), through funding provided by the Department of Pediatrics in the Duke University School of Medicine, and through funding from the National Institute of Allergy and Infectious Diseases (R01-AI161008 to M.S.K.). SARS-CoV-2 genome sequencing and analysis were supported by funding from the Office of the Provost at Duke University and the Duke Center for Genomic and Computational Biology. M.S.K. and J.H.H. were supported by the National Institutes of Health Career Development Awards (K23-AI135090 to M.S.K., K01-AI173398 to J.H.H.). The content is solely the responsibility of the authors and does not necessarily represent the official views of the National Institutes of Health. The funders had no role in study design, data collection and analysis, decision to publish, or preparation of the manuscript.

## Author contributions

Conceptualization: J.H.H., M.T.M., R.H., C.W.W., and M.S.K. Methodology: R.H., K.M.W., and M.S.K. Investigation: J.H.H., A.A.M., T.D., I.A.G., J.N.A., D.J.L., T.S.P., J.R., A.T.R., N.A.T., T.W.B., M.T.M., R.H., C.T.D., R.L., T.N.D., K.M.W., Z.X., A.M., O.R., C.W.W., and M.S.K. Visualization: A.A.M. and M.S.K. Funding acquisition: M.S.K. Project administration: J.H.H., C.W.W., and M.S.K. Supervision: M.S.K. Writing—original draft: J.H.H. and M.S.K. Writing—review & editing: J.H.H., A.A.M., T.D., I.A.G., J.N.A., D.J.L., T.S.P., J.R., A.T.R., N.A.T., T.W.B., M.T.M., R.H., C.T.D., R.L., T.N.D., K.M.W., Z.X., A.M., O.R., C.W.W., and M.S.K.

## Competing interests

T.W.B. is a consultant for and owns equity in Biomeme, Inc. K.M.W. held a sponsored research project from Moderna Therapeutics, Inc. on immune correlates of congenital CMV infection. C.W.W. is a consultant for and owns equity in Biomeme, Inc. M.S.K. is a consultant for Merck & Co, Inc. and Invivyd. N.A.T. has received research contracts with PDI, Purio, and Basilea as well as consulting for Techspert. A.M. is a consultant for Merck, Pfizer, Moderna, Astra-Zeneca, Enanta, and Sanofi-Pasteur. O.R. has received research grants from the Bill & Melinda Gates Foundation, Merck, and Janssen; fees for participation in advisory boards from Merck, Sanofi-Pasteur, Pfizer, and Moderna; and fees for lectures from Pfizer, AstraZeneca, Merck, and Sanofi-Pasteur. All other authors declare that they have no competing interests.

## Additional information

[1]Department of Pediatrics, Division of Infectious Diseases, Duke University School of Medicine, Durham, NC, USA. [2]Children's Health and Discovery Institute, Department of Pediatrics, Duke University School of Medicine, Durham, NC, USA. [3]Department of Biomedical Engineering, Duke University School of Medicine, Durham, NC, USA. [4]Department of Molecular Genetics and Microbiology, Duke University School of Medicine, Durham, NC, USA. [5]Duke University School of Medicine, Durham, NC, USA. [6]Children's Clinical Research Unit, Department of Pediatrics, Duke University School of Medicine, Durham, NC, USA. [7]Department of Pediatrics, Division of Pediatric Critical Care Medicine, Duke University School of Medicine, Durham, NC, USA. [8]Department of Medicine, Division of Infectious Diseases, Duke University School of Medicine, Durham, NC, USA. [9]Center for Infectious Disease Diagnostics and Innovation, Duke University School of Medicine, Durham, NC, USA. [10]Durham Veterans Affairs Medical Center, Durham, NC, USA. [11]Department of Biostatistics and Informatics, Duke University, Durham, NC, USA. [12]Duke Clinical Research Institute, Duke University School of Medicine, Durham, NC, USA. [13]Duke Human Vaccine Institute, Duke University School of Medicine, Durham, NC, USA. [14]Department of Neurosurgery, Duke University School of Medicine, Durham, NC, USA. [15]Department of Infectious Diseases, St. Jude Children's Research Hospital, Memphis, TN, USA. [16]These authors contributed equally: Jillian H. Hurst, Aditya A. Mohan. ✉e-mail: mkelly3@uams.edu

