## [Peer review file · Nature Communications]

Age-associated differences in mucosal and systemic host responses to SARS-CoV-2 infection

Corresponding Author: Dr Matthew Kelly

Version 0:

Reviewer comments:

Reviewer #1

(Remarks to the Author)

This manuscript by Hurst et al is interesting and well-written. It is certainly an interesting scientific question — to understand why COVID-19 presents differently in childhood; particularly as it is milder, in contrast to what we see to other respiratory infection. The methodology — bulk RNAseq — does present an opportunity to study this phenomenon at high-resolution. The actual analysis presented is quite superficial and more could have been done, but presumably these data will be made publicly available for other also to use this as a resource? The authors have done a reasonable job of explaining some of the limitations but the fact that the adults and child were recruited into different studies is potentially an important source of bias.

Further comments below:

Abstract

- The abstract seems too lengthy for Nat Comms.
- "Within individuals, robust URT immune responses to SARS-CoV-2 were associated with less peripheral immune activation, suggesting that local viral control may limit systemic responses." — This sentence ends with a discussion point. Abstracts typically stick to intro, methods, results and conclusions (rather than discussion)

Intro

- "This detailed analysis of host gene expression in one of the largest and most age-diverse cohorts studied to date reveals significant differences in the host response to SARS-CoV-2 infection across the lifespan." No need for the claim that this is one of the largest etc, just state what you did and what you found ...

Results

- "The prevalence of all evaluated symptoms but fever differed among SARS-CoV-2-infected subjects by age group (Table 1); these symptoms were generally more frequent among subjects 14-20 years of age (hereafter referred to as adolescents) and 21 years of age or older (adults) than among children 0-5 years of age (young children) and children 6-13 years of age (school-age children). — Unclear sentence needs rewriting."
- Characteristics of the study population, can you explain what the sampling protocol was, why and when, where and how they were individuals tested? This applies also for the adults. This is particularly important as adults and children recruited in different studies, so potential for bias ...
- Lines 176 and 179— "few genes", give exact numbers please.
- Line 207 — CIBERSORT is not directly measuring these cells so should be described as "suggested" increase in DC for example.
- Line 231 — This section stating a more robust URT response in younger individuals is based on higher normalised enrichment score in the younger infected vs healthy than the NES in the older groups. However, these this is not a direct comparison between the two ages groups, so the way the results are stated does not represent the analyses conducted.
- Line 249–251 — Interpretation of results is principally for the discussion rather than the results section, which should stick to objective reporting of the results.
- Line 266 — Similar to a previous comment the statement "adults tended to exhibit greater upregulation of these modules than pediatric age groups" was not demonstrated. There's a NES score for adults and a NES score for paediatrics age

groups. While the this is higher for adults, my understanding is you cannot say this represents a more generalisable finding of greater upregulation, without a formal statistical comparison? Much of this section has the same issue... This section (line 278–280) also ends with a comment that would be better in the Discussion section ...

- 291–292 — I'd prefer precision in this sentence rather than “many of these”... which?
- 299–301 — “Taken together, these data demonstrate that mucosal and systemic immune responses to acute SARS-CoV-2 infection are associated with, and likely contribute to, interindividual variability in the clinical manifestations of COVID-19.” — discussion not results
- In this final correlation analysis there seems to be conflicting findings. Interferon response in the upper respiratory tract correlate with peripheral blood but earlier you suggested more robust responses in the URT result in blunted peripheral blood responses?

Figure 1

You should mention of assumed bias in the cell proportions be mentioned. The “imputed immune cell proportions” will be odd for those familiar with the relative proportion of WBC — i.e. in blood neutrophils are the most abundant in humans but in figures show CD4s to the highest proportion with similar proportions of monocytes and neutrophils.

Reviewer #2

(Remarks to the Author)

This study uses a large number of samples collected from both pediatric and adult participants to characterise the transcriptional landscape of mucosal (in the case of pediatric) and systemic (pediatric and adult) immune responses to ancestral SARS-CoV-2 infection. The analysis is thorough, and the authors should be commended for the large number of samples collected.

Unfortunately, despite this, many of the findings observed in this paper have been shown previously in other studies, as the authors have mentioned in several sections. In many cases, the other studies have used more advanced technologies to assess both cellular and transcriptional responses (single-cell RNA sequencing combined with flow cytometry) and have been able to narrow down the responses to particular cell types of interest. It is also unclear how these findings from the ancestral strain in 2020 prior to variants of concern and global vaccine rollout are relevant for the current COVID-19 climate and literature.

There are several details of the cohorts that are missing in the methods and results that make the downstream assessment of findings difficult. Firstly, there is no description of the demographics of the healthy controls. Are these included in the total “n” numbers of table 1? The healthy controls should be listed separately within table 1 and a comparison of demographics between the SARS-CoV-2+ participants vs healthy participants performed. This is particularly relevant for Fig 2 which describes transcriptional differences between SARS-CoV-2+ and healthy. There are inconsistencies with the number of healthy controls reported, with the results stating n=49, but the abstract and introduction stating n=64.

The methods states that nasopharyngeal samples were collected from the adult participants however only peripheral blood findings are presented. Much later in the results, the authors elude that this is due to the incompatibility of the nasal samples for transcriptional analysis, but this should be stated early and included in the cohort description. This also means that the current manuscript title, highlighting that both mucosal and systemic immune responses were analysed in adults, is misleading and incorrect.

Were the adult blood samples collected and processed at the same centre using the same methods as the pediatric samples? If not, could any processing differences account for any of the downstream findings between pediatric and adult samples?

On lines 154-158, the authors state that viral loads were similar among SARS-CoV-2 infected subjects by age group and symptom presence. The authors then mention that only 28 NPA samples underwent SARS-CoV-2 testing. This naturally leads to the question about power for this analysis - what are the numbers in each age and symptom group for this analysis and is there power to make the statement that there are no differences?

Other comments:

The authors state in the introduction that the main advantage of their study over the previous similar studies is that they have been able to look at the different pediatric age groups (preschool, school age, adolescent). However, the majority of the figures (2,4,5,6) are not related to this comparison, and perform grouped analyses as per the other studies. There are other instances where interesting age-specific findings are moved to the supplement (such as those reported in lines 242-247) or it is unclear if a significant result was obtained (such as those reported in lines 266-268 and the use of the word “tended”). For all analyses that use CIBERSORT to report cell type proportions, it would be useful to compare the proportions identified by this method to other more gold standard methods such as flow cytometry or single-cell sequencing. There are several studies from children and adults that have used flow cytometry/scRNAseq on peripheral blood and nasal samples with publicly available data that can be used to determine the quality of the CIBERSORT analysis. This is particularly relevant for Figs 1 and 3 that rely on the CIBERSORT proportions for the biological comparisons.

Reviewer #3

(Remarks to the Author)

This manuscript describes immune populations and transcriptomes of individuals with acute SARS-CoV-2 infections and their uninfected contacts, with contrasts among healthy individuals between age groups; between infected and uninfected individuals to describe responses to acute infections; between symptomatic/asymptomatic (per symptom) infected individuals; and correlations drawn between baseline immune responses and convalescent neutralizing antibody titers. Despite its broad scope, the manuscript is clearly written and the data source described includes an age group of great

interest (young children <5). These are valuable data and I congratulate the authors on the hard work collecting them.

Overarching comments:

1. The largely-naive cohort is described as a strength, but I'm not sure this is actually a strength of the data. It's unclear how these results inform our understanding of the immune response to SARS-CoV-2 infections in a world with higher population immunity.
2. The lack of a comparator infection, and relatively brief discussion of findings from other infections (only one reference each regarding influenza and RSV) narrows the interpretation of these results. While the paper by Koch et al (reference 20) is limited by its heterogeneous and small cohort, these results (with RSV and influenza infections) merit further discussion/contrast to understand the specificity of the present results to SCV2.
3. The lack of an upper respiratory specimen from adults is a substantial limitation to the interpretation of age differences in the URT. Each of these contrasts, at most, compares young/school-aged children vs. adolescents, which makes extension of interpretation to broader age-related differences in severity more challenging.

Specific comments:

1. Counts appear to be off by 1. The authors, in multiple places, refer to a cohort of 201 individuals (137 infected/64 uninfected), but the results ("characteristics of the study population" section) refer to samples collected from 160 children and adolescents recruited through the BRAVE study (111 infected/49 uninfected) and an additional 42 non-hospitalized adults (26 infected/16 uninfected). In the "results" section, that adds up to 202 individuals (137 infected/65 uninfected). Double check.
2. Unclear in table 1 how many of these infections were medically attended or were infections detected among close contacts from research testing. It's also especially unclear if/how this varies by age. More adults were symptomatic; were they also more likely to be medically attended?
3. Unclear if symptom presence/absence was based on any of the timepoints assessed or only the acute timepoint.
4. Among uninfected participants, unclear the timing from exposure (or symptom onset in the infected household member) to specimen collection (other than "max 14 days"). This is especially important for understanding if the "uninfected" groups were just missed infections, since it seems like infections among contacts were determined based on a single swab taken up to 14 days after the index test. This would impact my interpretation of Figures 2 and 3, particularly in the youngest age group with only 10 "healthy" individuals.
5. Review captions and labelling for sub-panels b, c, and d in Figures 1 and 3.
6. In the results (Fig 1A and text): "We observed no significant differences in the proportions of immune cells within the URT of healthy children and adolescents by age (Fig. 1a)." These differences were only observed in the population-adjusted expression analyses, in both the differential gene expression in Fig S1A and in the "modules" described in Fig 1c. On a statistical level, it's unclear if these differences would be significant without accounting for the proportions of immune cells (which are better supported in peripheral blood analyses). On an interpretation level, I'm not clear how this nuance relates to the language in the final sentence of results paragraph 3 ("greater activation of immune cell populations in the URT, which may prime children for..." or the language in the discussion about greater "URT immune tone".
7. Figure 4/the analysis of symptoms would especially benefit from nuanced engagement with the "medically attended vs. contact" distinction.
8. In figure 6, there are two observations with high convalescent serum neutralizing activity, who appear to be influencing at least the strength of the associations in this pediatric sample. Are these individuals otherwise outliers in e.g. time to specimen collection (with an IQR of 52-87 days, not sure the range here)?

Version 1:

Reviewer comments:

Reviewer #1

(Remarks to the Author)

I am satisfied with the revisions that were made on the manuscript.

Editorial note

This reviewer was additionally asked to comment in the place of reviewer 2 and 3 who were unable to provide a response in this round of review.

I've had a look at this and my view is the authors have adequately addressed the reviewers' comments.

(Remarks on code availability)

Reviewer #4

(Remarks to the Author)

Regarding the response to reviewer #2, the authors sufficiently highlighted the aspects analyzed in this study, particularly the opportunity to understand innate and adaptive immune pathways in the UTR in a SARS-CoV-2 naive pediatric and adult population. This is a large cohort which targets the early phase of the pandemic, although not generalizable to the situation we have now with different history of vaccination and contact to circulating VOCs such as omicron, however, the study constitutes an important contribution to understand viral infection by SARS-CoV-2 in general and to allow future analogous deductions from this findings in the case new viral threats emerge.

Additions by table S1 are helpful to understand the cohort analyzed in the study. All of the major comments regarding illustration of the data and showing data on both pediatric and adult groups as well as giving details about the methods used and handling of samples have been addressed in the revised manuscript.

Of course, single cell sequencing and other modern methods may have elucidate more details on the single cell pathways, however, considering the important contribution by analyzing a pediatric and adult SARS-CoV-2-naive populations at their first contacts to SARS-CoV-2 is worth to be published as analogical conclusions may be derived from this study in the case new viral pathogens emerge.

Regarding reviewer #3 I agree that the cohort is rather small to completely understand the anti-viral mechanisms in children and adults. However, the data seem worth to be published as they represent a real-world situation with high inter-individual variance. The concerns of reviewer #3 have been accordingly addressed by the authors and included into the discussion section. Limitations of the study particularly regarding the methods and the generalizibility of results to the current situation are sufficiently discussed in the revised manuscript.

(Remarks on code availability)

Regarding the response to reviewer #2, the authors sufficiently highlighted the aspects analyzed in this study, particularly the opportunity to understand innate and adaptive immune pathways in the UTR in a SARS-CoV-2 naive pediatric and adult population. This is a large cohort which targets the early phase of the pandemic, although not generalizable to the situation we have now with different history of vaccination and contact to circulating VOCs such as omicron, however, the study constitutes an important contribution to understand viral infection by SARS-CoV-2 in general and to allow future analogous deductions from this findings in the case new viral threats emerge.

Additions by table S1 are helpful to understand the cohort analyzed in the study. All of the major comments regarding illustration of the data and showing data on both pediatric and adult groups as well as giving details about the methods used and handling of samples have been addressed in the revised manuscript.

Of course, single cell sequencing and other modern methods may have elucidate more details on the single cell pathways, however, considering the important contribution by analyzing a pediatric and adult SARS-CoV-2-naive populations at their first contacts to SARS-CoV-2 is worth to be published as analogical conclusions may be derived from this study in the case new viral pathogens emerge.

Regarding reviewer #3 I agree that the cohort is rather small to completely understand the anti-viral mechanisms in children and adults. However, the data seem worth to be published as they represent a real-world situation with high inter-individual variance. The concerns of reviewer #3 have been accordingly addressed by the authors and included into the discussion section. Limitations of the study particularly regarding the methods and the generalizibility of results to the current situation are sufficiently discussed in the revised manuscript.

Response to Reviewers

Reviewer #1 (Remarks to the Author):

This manuscript by Hurst et al is interesting and well-written. It is certainly an interesting scientific question — to understand why COVID-19 presents differently in childhood; particularly as it is milder, in contrast to what we see to other respiratory infection. The methodology — bulk RNAseq — does present an opportunity to study this phenomenon at high-resolution. The actual analysis presented is quite superficial and more could have been done, but presumably these data will be made publicly available for other also to use this as a resource? The authors have done a reasonable job of explaining some of the limitations but the fact that the adults and child were recruited into different studies is potentially an important source of bias.

We thank the reviewer for these helpful comments. Of note, all de-identified, individual-level raw data, clinical metadata, and associated analysis scripts have been made publicly available in the Gene Expression Omnibus, Sequence Read Archive, and on github. We certainly hope that these data can be used by other investigators to further advance our understanding of host factors that contribute to variability in the outcomes of SARS-CoV-2 infection. We have also revised the Results (Pages 6-7) and Methods (Pages 22-24) sections to clarify the study procedures and to provide additional details regarding how samples were collected and processed for these studies. We believe this additional information provides further support that our findings are unlikely to be attributable to differences in study procedures, including the procedures used for sample collection and processing.

Further comments below:

Abstract

- **The abstract seems too lengthy for Nat Comms.**

We agree with the reviewer and have shortened the abstract to 150 words to meet the publication guidelines (page 3).

Abstract

Age is among the strongest risk factors for severe outcomes from SARS-CoV-2 infection. We profiled upper respiratory tract (URT) and peripheral blood transcriptomes of 202 participants (age range of 1 week to 83 years), including 137 non-hospitalized individuals with mild SARS-CoV-2 infection and 65 healthy individuals. Among healthy children and adolescents, younger age was associated with higher URT expression of innate and adaptive immune pathways. SARS-CoV-2 infection induced broad upregulation of URT innate and adaptive immune responses among children and adolescents. Peripheral blood responses among SARS-CoV-2-infected children and adolescents were dominated by interferon pathways, while upregulation of myeloid activation, inflammatory, and coagulation pathways was observed only in adults. Among SARS-CoV-2-infected individuals, fever was associated with blunted URT immune responses and more pronounced systemic immune activation. These findings demonstrate that immune responses to SARS-CoV-2 differ across the lifespan, including during childhood and adolescence, and contribute to age-associated differences in COVID-19 presentation.

- **"Within individuals, robust URT immune responses to SARS-CoV-2 were associated with less peripheral immune activation, suggesting that local viral control may limit systemic responses." — This sentence ends with a discussion point. Abstracts typically stick to intro, methods, results and conclusions (rather than discussion)**

We thank the reviewer for this helpful comment. This line has been removed from the Abstract.

Intro

- **"This detailed analysis of host gene expression in one of the largest and most age-diverse cohorts studied to date reveals significant differences in the host response to SARS-CoV-2 infection across the lifespan." No need for the claim that this is one of the largest etc, just state what you did and what you found ...**

We have removed this statement from the Introduction.

Results

- **"The prevalence of all evaluated symptoms but fever differed among SARS-CoV-2-infected subjects by age group (Table 1); these symptoms were generally more frequent among subjects 14-20 years of age (hereafter referred to as adolescents) and 21 years of age or older (adults) than among children 0-5 years of age (young children) and children 6-13 years of age (school-age children). — Unclear sentence needs rewriting."**

We thank the reviewer for this comment, and have revised this text in the manuscript as follows:

Page 8, Lines 148-154: *"The prevalence of most symptoms differed by age among SARS-CoV-2-infected participants (Table 1). Compared to young children and school-age children, adolescents and adults more frequently reported cough (58% vs. 24%; Chi-square test, $p < 0.0001$), headache (52% vs. 15%; Chi-square test, $p < 0.0001$), myalgias (43% vs. 13%, Chi-square test, $p = 0.0001$), nasal congestion (38% vs. 7%, Chi-square test, $p < 0.0001$), rhinorrhea (35% vs. 7%, Chi-square test, $p < 0.0001$), loss of smell (45% vs. 6%; Fisher's exact test, $p < 0.0001$), and loss of taste (38% vs. 4%; Fisher's exact test, $p < 0.0001$)."*

- **Characteristics of the study population, can you explain what the sampling protocol was, why and when, where and how they were individuals tested? This applies also for the adults. This is particularly important as adults and children recruited in different studies, so potential for bias ...**

We thank the reviewer for this helpful suggestion. The BRAVE Kids study (children and adolescents <21 years of age) was designed in consultation with the investigators of the MESSI study (adults ≥ 21 years), the latter of which was recruiting individuals with community-acquired infections prior to the COVID-19 pandemic. During this process, procedures for recruitment and sample collection and processing were aligned across these studies to the extent possible. The two study teams frequently conducted joint home visits, with the BRAVE Kids study team enrolling children and adolescents in a household and the MESSI study team enrolling adults, and the samples collected during these visits were processed by the same core facility at Duke using established facility protocols. We additionally ensured that all samples underwent RNA extraction and library preparation using the same sample workflow and, where possible, sequencing batches included samples from both studies.

Of note, both studies focused on two populations: individuals who were tested for SARS-CoV-2 in an outpatient clinical setting, and individuals who had a close contact (typically a household member) who had tested positive for SARS-CoV-2 by PCR. At the time of this study, the vast majority of the testing performed locally was conducted at drive-up testing sites. Individuals who wished to be tested generally needed to have a telehealth visit with their provider to determine if they met guidelines for receipt of SARS-CoV-2 testing, and were subsequently referred to a testing site. Testing guidelines at the time required the presence of one or more COVID-19 symptoms and/or known close contact with an individual with confirmed SARS-CoV-2 infection. We additionally found that a subset of participants – most often those who did not have an established medical home – presented for urgent or emergency care to access SARS-CoV-2 testing, even in the absence of symptoms.

We have expanded relevant sections in the Results (Pages 6-7) and Methods (Pages 22-24) sections to clarify the study procedures and to provide additional details regarding how samples were collected and processed for these studies:

Results

Characteristics of the study population

The samples and data included in this analysis were collected from participants in two studies conducted within the Duke University Health System (DUHS): the Biorepository of Respiratory Virus Exposed (BRAVE) Kids study, which recruited SARS-CoV-2-exposed children and adolescents less than 21 years of age, and the Molecular and Epidemiological Study of Suspected Infection (MESSI) study, which recruited SARS-CoV-2-exposed adults 21 years of age or older (Table 1, Table S1). Recruitment in both studies included non-hospitalized participants who presented for SARS-CoV-2 testing within the health system and/or who had known close contact with an individual with confirmed SARS-CoV-2 infection (typically a household member). Participants in both studies were identified through review of SARS-CoV-2 testing conducted in the DUHS, and the study teams additionally approached close contacts of index cases for study participation. All participants included in this analysis were recruited between April 1, 2020, and December 31, 2020 (7). None of the participants had a known SARS-CoV-2 infection prior to the current illness, nor had participants received a COVID-19 vaccine at the time of enrollment.

Methods

Study Design

The children and adolescents (<21 years) included in these analyses were enrolled in the Biospecimens from Respiratory Virus-Exposed Kids (BRAVE Kids) study, a prospective cohort study of non-hospitalized individuals with confirmed SARS-CoV-2 infection or close contact with an individual with confirmed infection (7). Participants were identified either through presentation to the health system for SARS-CoV-2 testing or through identification of a close contact with PCR-confirmed SARS-CoV-2 infection. At the time of enrollment, a study team member administered a questionnaire by telephone to the participant or a caregiver to gather information on sociodemographic factors, potential sources of SARS-CoV-2 exposure, and relevant past medical history. Questionnaires were also administered to assess the presence and duration of specific symptoms at enrollment and at 7, 14, and 28 days after enrollment or until participants reported resolution of all symptoms. Nasopharyngeal samples were collected with nylon flocked swabs (Copan Italia, Brescia, Italy), placed into RNAProtect (Qiagen, Hilden, Germany), and tested for SARS-CoV-2 by quantitative PCR (7). Whole blood samples were collected into PAXgene blood RNA tubes (Qiagen).

The adults (≥21 years) included in these analyses participated in the Molecular and Epidemiological Study of Suspected Infection (MESSI) study, a prospective cohort study of individuals with confirmed SARS-CoV-2 infection or close contact with an individual with confirmed infection. Although this study additionally enrolled individuals hospitalized for COVID-19, only non-hospitalized individuals were included in the present analyses. Nasopharyngeal samples were collected using prepackaged kits containing nylon flocked swabs and viral transport medium (VTM; Dasky Life Science, Ningbo, China). Whole blood samples were collected into PAXgene blood RNA tubes (Qiagen). Detailed data on symptoms, exposures, and medical history were similarly collected from MESSI study participants using serial questionnaires.

While the children and adults in this analysis were enrolled into separate study protocols, the two study teams coordinated study activities to the extent possible, frequently conducting joint visits to households with multiple eligible participants and using the same sample collection and processing procedures. The primary difference in the procedures of these two studies was in the medium used for nasopharyngeal swab samples; samples in the BRAVE Kids study were collected into RNAProtect and samples in the MESSI study were collected into VTM, with the latter medium precluding transcriptomic analyses. Nasopharyngeal and blood samples from both studies were initially processed and stored in the Duke Human Vaccine Institute Accessioning Unit. These samples underwent RNA extraction and library preparation using the same workflow in the Duke University Sequencing and Genomic Technologies core facility. Finally, whenever possible, samples from both studies were included in sequencing batches to ensure that results could be compared across studies.

• **Lines 176 and 179— “few genes” , give exact numbers please.**

We have revised these lines to provide the exact numbers of differentially expressed genes.

Page 9, Lines 181-185: “*Within the URT of healthy subjects, we identified 5 genes that were differentially expressed between young children and school-age children and 2 genes that were differentially expressed between school-age children and adolescents (Fig. S1a). In contrast, we identified 36 genes that were differentially expressed in young children compared to adolescents.*”

• **Line 207 — CIBERSORT is not directly measuring these cells so should be described as “suggested” increase in DC for example.**

We agree with the reviewer and have modified references to CIBERSORT to clarify that these are only suggested cell proportions.

Page 9, Lines 168-176: “*Because immune cell composition is known to vary with age (24), we estimated the proportions of immune cell populations within URT and peripheral blood samples using the CIBERSORT deconvolution algorithm and the LM22 signature matrix (25). We observed no significant differences in the imputed proportions of immune cells within the URT of healthy children and adolescents by age (Fig. 1a). Within the peripheral blood of children, adolescents, and adults, we found that the proportions of transcripts associated with B cells (beta regression, $p_{adj}<0.0001$), CD8+ T cells ($p_{adj}=0.002$), and plasma cells ($p_{adj}=0.005$) decreased with increasing age, while the proportions of transcripts associated with monocytes and macrophages ($p_{adj}<0.0001$) and neutrophils ($p_{adj}=0.0009$) increased (Fig. 1b).*”

• **Line 231 — This section stating a more robust URT response in younger individuals is based on higher normalised enrichment score in the younger infected vs healthy than the NES in the older groups. However, these this is not a direct comparison between the two ages groups, so the way the results are stated does not represent the analyses conducted.**

We agree with the reviewer that the phrasing of these statements did not fully convey these comparisons, and we have revised these sentences to more accurately reflect the performed analyses. Moreover, using enrichment scores from single-sample gene set enrichment analysis (ssGSEA) as the outcome variables, we now fit linear regression models with an interaction term between SARS-CoV-2 infection status and age group. This approach enables us to directly compare the degree of change of expression of immune modules associated with SARS-CoV-2 infection across age groups.

Page 12, Lines 250-258: “*Finally, to directly compare URT responses to SARS-CoV-2 across pediatric age groups, we used single-sample gene set enrichment analysis (ssGSEA) to generate enrichment scores corresponding to the levels of expression of immune modules within each sample. We then fit linear regression models with module enrichment scores as the dependent variables and included an interaction term between SARS-CoV-2 infection status and age group in these models, enabling direct comparison of changes in immune module expression associated with SARS-CoV-2 infection by age group. Using this approach, we did not observe any significant differences in the degree to which these modules were upregulated within the URT in association with SARS-CoV-2 infection across pediatric age groups (Fig. 3c).*”

• **Line 249–251 — Interpretation of results is principally for the discussion rather than the results section, which should stick to objective reporting of the results.**

We have removed these lines from the Results and revisit this topic in the Discussion.

• **Line 266 — Similar to a previous comment the statement “adults tended to exhibit greater upregulation of these modules than pediatric age groups ” was not demonstrated. There’s a NES score for adults and a NES score for paediatrics age groups. While the this is higher for adults, my understanding is you cannot say this represents a more generalisable finding of greater upregulation,**

without a formal statistical comparison? Much of this section has the same issue... This section (line 278–280) also ends with a comment that would be better in the Discussion section ...

We thank the reviewer for these helpful suggestions and have worked to clarify the comparisons that were made in these analyses. As described above for the mucosal (URT) immune responses, we now present the results of linear regression models incorporating an interaction term between SARS-CoV-2 infection status and age group, enabling a direct comparison of the changes in immune module expression associated with SARS-CoV-2 across age groups. We now only highlight those comparisons that were statistically significant ($p < 0.05$) or provide p -values for nonsignificant results.

Page 13, Lines 284-297: “We then used ssGSEA and linear regression to compare the systemic immune responses associated with SARS-CoV-2 infection across age groups. Compared to other pediatric age groups and adults, young children had less systemic upregulation of genes corresponding to myeloid activation (linear regression; vs. school-age children: $p=0.04$, vs. adolescents: $p=0.02$, vs. adults: $p=0.048$), phagocytosis (vs. school-age children: $p=0.03$, vs. adolescents: $p=0.0005$, vs. adults: $p=0.07$), and lymphocyte trafficking (vs. school-age children: $p=0.002$, vs. adolescents: $p=0.008$, vs. adults: $p=0.03$) associated with SARS-CoV-2 infection. Comparing adults to pediatric age groups, SARS-CoV-2 infection induced greater peripheral blood upregulation of genes involved with the complement system (vs. young children: $p=0.01$, vs. school-age children: $p=0.051$, vs. adolescents, $p=0.14$), and less upregulation of genes corresponding to T cell receptor signaling (vs. young children, $p=0.01$, vs. school-age children, $p=0.04$, vs. adolescents, $p=0.002$) and regulatory T cell differentiation (vs. young children, $p=0.04$, vs. school-age children, $p=0.08$, vs. adolescents, $p=0.009$). These findings demonstrate that the systemic immune responses to SARS-CoV-2 infection differ across the age spectrum, including between pediatric age groups.”

• 291–292 — I’d prefer precision in this sentence rather than “many of these”... which?

We agree with the reviewer and have revised this section to specify immune modules of interest.

Page 15, Lines 306-318: “Participants who reported fever exhibited downregulation of innate immune (e.g., interferon signaling, TNF signaling, Nod-like receptor signaling, RNA sensing, NK cell activity) and adaptive immune pathways (e.g., MHC class I/II presentation, T and B cell receptor signaling, NF- κ B signaling, JAK/STAT signaling, lymphocyte trafficking, immune memory and immune exhaustion) in the URT, and upregulation of many of these same pathways (e.g., interferon signaling, NK cell activity, MHC class I, T cell receptor signaling, NF- κ B signaling, and lymphocyte trafficking) in peripheral blood. In contrast, participants who reported cough or headache exhibited broad upregulation of innate and adaptive immune signaling pathways in both the URT and peripheral blood, including modules corresponding to innate immune cell activation, interferon signaling, phagocytosis, myeloid cell activation, and MHC class I presentation. Participants reporting symptoms localizing to the URT, including rhinorrhea, nasal congestion, and loss of smell or taste, exhibited broad downregulation of innate and adaptive immune responses in the URT and downregulation of interferon signaling pathways in peripheral blood.”

• 299–301 — “Taken together, these data demonstrate that mucosal and systemic immune responses to acute SARS-CoV-2 infection are associated with, and likely contribute to, interindividual variability in the clinical manifestations of COVID-19.” — discussion not results

We agree with the reviewer and have removed these lines.

• In this final correlation analysis there seems to be conflicting findings. Interferon response in the upper respiratory tract correlate with peripheral blood but earlier you suggested more robust responses in the URT result in blunted peripheral blood responses?

We thank the reviewer for pointing out this potentially confusing interpretation of these data. Note that we have revised the manuscript to indicate that SARS-CoV-2 infection induced broad upregulation of innate and adaptive immune modules within the URT, with no statistically significant differences in the magnitude

of these responses across pediatric age groups. Using enrichment scores from single-sample gene set enrichment analysis (ssGSEA) as the outcome variables, we now fit linear regression models with an interaction term between SARS-CoV-2 infection status and age group. This approach enables us to directly compare the degree of change of expression of immune modules associated with SARS-CoV-2 infection across age groups.

Page 12, Lines 250-258: “*Finally, to directly compare URT responses to SARS-CoV-2 across pediatric age groups, we used single-sample gene set enrichment analysis (ssGSEA) to generate enrichment scores corresponding to the levels of expression of immune modules within each sample. We then fit linear regression models with module enrichment scores as the dependent variables and included an interaction term between SARS-CoV-2 infection status and age group in these models, enabling direct comparison of changes in immune module expression associated with SARS-CoV-2 infection by age group. Using this approach, we did not observe any significant differences in the degree to which these modules were upregulated within the URT in association with SARS-CoV-2 infection across pediatric age groups (Fig. 3c).*”

Moreover, we would like to point out that all analyses presented prior to those corresponding to Figure 5 were conducted across age groups or comparing SARS-CoV-2-infected and uninfected individuals. In contrast, in the analyses referenced by the reviewer in this comment, we evaluated correlations between expression of immune modules in the URT and peripheral blood *within* individual participants. Although these analyses suggest that the expression of interferon-associated modules in the URT is positively correlated with expression of these modules in peripheral blood, expression of interferon modules in the URT is negatively correlated with peripheral blood expression of several other immune modules. Moreover, as is shown in the correlation matrix in Figure 5, a number of other negative correlations were observed between the expression of innate and adaptive immune modules in the URT and the expression of (primarily innate) immune modules in peripheral blood.

Figure 1

You should mention of assumed bias in the cell proportions be mentioned. The “imputed immune cell proportions” will be odd for those familiar with the relative proportion of WBC — i.e. in blood neutrophils are the most abundant in humans but in figures show CD4s to the highest proportion with similar proportions of monocytes and neutrophils.

We thank the reviewer for this helpful suggestion. We have revised the figure legend as follows:

Page 40, Lines 922-925: “*Box and whisker plots depict proportions of transcripts attributed to different immune cell populations in the upper respiratory tract and peripheral blood of healthy children and adolescents, with cell proportions imputed using CIBERSORT. Note that the proportions of cell types predicted by CIBERSORT do not reflect their absolute proportions within a given sample type.*”

Additionally, we now clarify this point in the description of CIBERSORT in the Methods section and in the legends for Figure 3 and Supplementary Figure 2.

Reviewer #2 (Remarks to the Author):

This study uses a large number of samples collected from both pediatric and adult participants to characterise the transcriptional landscape of mucosal (in the case of pediatric) and systemic (pediatric and adult) immune responses to ancestral SARS-CoV-2 infection. The analysis is thorough, and the authors should be commended for the large number of samples collected.

Unfortunately, despite this, many of the findings observed in this paper have been shown previously in other studies, as the authors have mentioned in several sections. In many cases, the other studies have used more advanced technologies to assess both cellular and transcriptional responses (single-cell RNA sequencing combined with flow cytometry) and have been able to narrow down the responses to particular cell types of interest. It is also unclear how these findings from the ancestral

strain in 2020 prior to variants of concern and global vaccine rollout are relevant for the current COVID-19 climate and literature.

We thank the reviewer for these helpful comments. We have revised the Results (Pages 6-7) and Methods (Pages 22-24) sections to clarify how participants were recruited into these studies and how samples were collected and processed.

Results

Characteristics of the study population

The samples and data included in this analysis were collected from participants in two studies conducted within the Duke University Health System (DUHS): the Biorepository of Respiratory Virus Exposed (BRAVE) Kids study, which recruited SARS-CoV-2-exposed children and adolescents less than 21 years of age, and the Molecular and Epidemiological Study of Suspected Infection (MESSI) study, which recruited SARS-CoV-2-exposed adults 21 years of age or older (Table 1, Table S1). Recruitment in both studies included non-hospitalized participants who presented for SARS-CoV-2 testing within the health system and/or who had known close contact with an individual with confirmed SARS-CoV-2 infection (typically a household member). Participants in both studies were identified through review of SARS-CoV-2 testing conducted in the DUHS, and the study teams additionally approached close contacts of index cases for study participation. All participants included in this analysis were recruited between April 1, 2020, and December 31, 2020 (7). None of the participants had a known SARS-CoV-2 infection prior to the current illness, nor had participants received a COVID-19 vaccine at the time of enrollment.

Methods

Study Design

The children and adolescents (<21 years) included in these analyses were enrolled in the Biospecimens from Respiratory Virus-Exposed Kids (BRAVE Kids) study, a prospective cohort study of non-hospitalized individuals with confirmed SARS-CoV-2 infection or close contact with an individual with confirmed infection (7). Participants were identified either through presentation to the health system for SARS-CoV-2 testing or through identification of a close contact with PCR-confirmed SARS-CoV-2 infection. At the time of enrollment, a study team member administered a questionnaire by telephone to the participant or a caregiver to gather information on sociodemographic factors, potential sources of SARS-CoV-2 exposure, and relevant past medical history. Questionnaires were also administered to assess the presence and duration of specific symptoms at enrollment and at 7, 14, and 28 days after enrollment or until participants reported resolution of all symptoms. Nasopharyngeal samples were collected with nylon flocked swabs (Copan Italia, Brescia, Italy), placed into RNAprotect (Qiagen, Hilden, Germany), and tested for SARS-CoV-2 by quantitative PCR (7). Whole blood samples were collected into PAXgene blood RNA tubes (Qiagen).

The adults (≥21 years) included in these analyses participated in the Molecular and Epidemiological Study of Suspected Infection (MESSI) study, a prospective cohort study of individuals with confirmed SARS-CoV-2 infection or close contact with an individual with confirmed infection. Although this study additionally enrolled individuals hospitalized for COVID-19, only non-hospitalized individuals were included in the present analyses. Nasopharyngeal samples were collected using prepackaged kits containing nylon flocked swabs and viral transport medium (VTM; Dasky Life Science, Ningbo, China). Whole blood samples were collected into PAXgene blood RNA tubes (Qiagen). Detailed data on symptoms, exposures, and medical history were similarly collected from MESSI study participants using serial questionnaires.

While the children and adults in this analysis were enrolled into separate study protocols, the two study teams coordinated study activities to the extent possible, frequently conducting joint visits to households with multiple eligible participants and using the same sample collection and processing procedures. The primary difference in the procedures of these two studies was in the medium used for nasopharyngeal swab

samples; samples in the BRAVE Kids study were collected into RNAProtect and samples in the MESSI study were collected into VTM, with the latter medium precluding transcriptomic analyses. Nasopharyngeal and blood samples from both studies were initially processed and stored in the Duke Human Vaccine Institute Accessioning Unit. These samples underwent RNA extraction and library preparation using the same workflow in the Duke University Sequencing and Genomic Technologies core facility. Finally, whenever possible, samples from both studies were included in sequencing batches to ensure that results could be compared across studies.

We additionally have revised the Introduction and Discussion to describe an important aspect of conducting these analyses on samples from individuals infected early in the COVID-19 pandemic with an ancestral SARS-CoV-2 strain:

Page 5, Lines 96-99: *“SARS-CoV-2-infected participants were enrolled early in the COVID-19 pandemic, prior to the widespread circulation of major variants of concern and the routine availability of COVID-19 vaccines; thus, this analysis focused on immune responses to SARS-CoV-2 among a population that was naïve to the virus.”*

Pages 21-22, Lines 461-466: *“Finally, this analysis focused on participants infected with SARS-CoV-2 early in the pandemic, which may limit the generalizability of our findings to infections caused by currently circulating SARS-CoV-2 variants in a population with substantial herd immunity. Conversely, this SARS-CoV-2-naïve population presents a unique opportunity to understand how individuals respond to SARS-CoV-2 infection in the absence of prior immune experience, and may also provide important insights into how age influences responses to other emerging viral pathogens.”*

There are several details of the cohorts that are missing in the methods and results that make the downstream assessment of findings difficult. Firstly, there is no description of the demographics of the healthy controls. Are these included in the total “n” numbers of table 1? The healthy controls should be listed separately within table 1 and a comparison of demographics between the SARS-CoV-2+ participants vs healthy participants performed. This is particularly relevant for Fig 2 which describes transcriptional differences between SARS-CoV-2+ and healthy. There are inconsistencies with the number of healthy controls reported, with the results stating n=49, but the abstract and introduction stating n=64.

We agree with the reviewer that additional details about the cohort would aid in the interpretation of the manuscript. The demographics of the healthy controls were included in Table 1. We have now created a supplemental table (Table S1) that directly compares the age, sex, and prevalence of comorbidities across age groups. We have additionally revised the Results to describe these data:

Page 8, Lines 154-157: *“Within each age group, there were no differences in the age, sex, or prevalence of obesity among SARS-CoV-2-infected and uninfected individuals (**Table S1**); however, the prevalence of other comorbidities among infected adults was higher than among uninfected adults (64% vs. 19%; Fisher’s exact test, $p=0.009$).”*

The methods states that nasopharyngeal samples were collected from the adult participants however only peripheral blood findings are presented. Much later in the results, the authors elude that this is due to the incompatibility of the nasal samples for transcriptional analysis, but this should be stated early and included in the cohort description. This also means that the current manuscript title, highlighting that both mucosal and systemic immune responses were analysed in adults, is misleading and incorrect.

We thank the reviewer for bringing up this important point. We have modified the manuscript title to the following: “Age-associated differences in mucosal and systemic host responses to SARS-CoV-2 infection”.

Were the adult blood samples collected and processed at the same centre using the same methods as the pediatric samples? If not, could any processing differences account for any of the downstream findings between pediatric and adult samples?

We now provide additional information on how the samples were collected and processed in these studies, as described previously in this document and as presented in the Results (Pages 6-7) and Methods (Pages 22-24) sections of the revised manuscript. Given the extent to which the procedures for sample collection and processing, RNA extraction, and library preparation were aligned across these studies, as well as the inclusion of samples from both studies in sequencing batches whenever possible, we do not believe that our findings are attributable to differences in sample processing.

On lines 154-158, the authors state that viral loads were similar among SARS-CoV-2 infected subjects by age group and symptom presence. The authors then mention that only 28 NPA samples underwent SARS-CoV-2 testing. This naturally leads to the question about power for this analysis - what are the numbers in each age and symptom group for this analysis and is there power to make the statement that there are no differences?

We have provided further clarifications regarding the numbers of participants who provided data for these analyses. Note that nasopharyngeal SARS-CoV-2 viral loads were measured by qPCR for the vast majority of SARS-CoV-2-infected individuals (115 of 137, 84%). In contrast, SARS-CoV-2 genomic sequencing was performed on a subset of 28 samples across the collection period in order to provide some insight into which viral lineages were present among infected participants. We have revised this section as follows:

Page 8, Lines 157-163: “Nasopharyngeal SARS-CoV-2 viral loads were also similar among SARS-CoV-2-infected subjects by age group (young children: n=26, school-age children: n= 31, adolescents: n=33, adults: n=25; Kruskal-Wallis test, p=0.37) and symptom presence (symptomatic subjects: n=72, asymptomatic subjects: n=43; Wilcoxon rank-sum test, p=0.37). A subset of nasopharyngeal samples from SARS-CoV-2-infected participants (n=28) underwent genomic sequencing to identify the infecting lineage, with only ancestral strains being identified in these individuals (Table S2).”

Other comments:

The authors state in the introduction that the main advantage of their study over the previous similar studies is that they have been able to look at the different pediatric age groups (preschool, school age, adolescent). However, the majority of the figures (2,4,5,6) are not related to this comparison, and perform grouped analyses as per the other studies. There are other instances where interesting age-specific findings are moved to the supplement (such as those reported in lines 242-247) or it is unclear if a significant result was obtained (such as those reported in lines 266-268 and the use of the word “tended”).

We appreciate the reviewer’s feedback and have modified the manuscript to more clearly present differences in SARS-CoV-2-associated immune responses by age group. Further, we have revised the statistical analysis to permit direct comparisons of the immune responses associated with SARS-CoV-2 infection across age groups. Using enrichment scores from single-sample gene set enrichment analysis (ssGSEA) as the outcome variables, we now fit linear regression models with an interaction term between SARS-CoV-2 infection status and age group. This approach enables us to directly compare the degree of change of expression of URT and peripheral blood immune modules associated with SARS-CoV-2 infection across age groups. The results of these analyses are presented in the below paragraphs in the Results section of the revised manuscript. Additionally, throughout the Results section, we now present only statistically significant differences in comparisons of immune responses associated with SARS-CoV-2 infection across age groups (or provide p-values for nonsignificant results).

Page 12, Lines 250-258: “Finally, to directly compare URT responses to SARS-CoV-2 across pediatric age groups, we used single-sample gene set enrichment analysis (ssGSEA) to generate enrichment scores corresponding to the levels of expression of immune modules within each sample. We then fit linear

regression models with module enrichment scores as the dependent variables and included an interaction term between SARS-CoV-2 infection status and age group in these models, enabling direct comparison of changes in immune module expression associated with SARS-CoV-2 infection by age group. Using this approach, we did not observe any significant differences in the degree to which these modules were upregulated within the URT in association with SARS-CoV-2 infection across pediatric age groups (Fig. 3c)."

Page 13, Lines 284-297: *"We then used ssGSEA and linear regression to compare the systemic immune responses associated with SARS-CoV-2 infection across age groups. Compared to other pediatric age groups and adults, young children had less systemic upregulation of genes corresponding to myeloid activation (linear regression; vs. school-age children: $p=0.04$, vs. adolescents: $p=0.02$, vs. adults: $p=0.048$), phagocytosis (vs. school-age children: $p=0.03$, vs. adolescents: $p=0.0005$, vs. adults: $p=0.07$), and lymphocyte trafficking (vs. school-age children: $p=0.002$, vs. adolescents: $p=0.008$, vs. adults: $p=0.03$) associated with SARS-CoV-2 infection. Comparing adults to pediatric age groups, SARS-CoV-2 infection induced greater peripheral blood upregulation of genes involved with the complement system (vs. young children: $p=0.01$, vs. school-age children: $p=0.051$, vs. adolescents, $p=0.14$), and less upregulation of genes corresponding to T cell receptor signaling (vs. young children, $p=0.01$, vs. school-age children, $p=0.04$, vs. adolescents, $p=0.002$) and regulatory T cell differentiation (vs. young children, $p=0.04$, vs. school-age children, $p=0.08$, vs. adolescents, $p=0.009$). These findings demonstrate that the systemic immune responses to SARS-CoV-2 infection differ across the age spectrum, including between pediatric age groups."*

For all analyses that use CIBERSORT to report cell type proportions, it would be useful to compare the proportions identified by this method to other more gold standard methods such as flow cytometry or single-cell sequencing. There are several studies from children and adults that have used flow cytometry/scRNAseq on peripheral blood and nasal samples with publicly available data that can be used to determine the quality of the CIBERSORT analysis. This is particularly relevant for Figs 1 and 3 that rely on the CIBERSORT proportions for the biological comparisons.

We thank the reviewer for these important comments. Because we identified few differences in imputed immune cell proportions by age in nasopharyngeal samples, and CIBERSORT has not been formally validated for analyses of this sample type, we present immune cell proportions for nasopharyngeal samples but no longer adjust analyses of upper respiratory gene expression for immune cell composition. Note that this modification did not substantively change the results of these analyses. Additionally, the original description of CIBERSORT validated its use in analyses of immune cell populations in whole blood samples using the same signature matrix (LM22) that we used in our analysis and flow cytometry data as the reference (PMC4739640). Given the prior validation of this approach for whole blood samples, and the establishes differences in peripheral blood immune cell composition across the age spectrum, we continue to present analyses of peripheral blood gene expression adjusted for immune cell proportions.

Reviewer #3 (Remarks to the Author):

This manuscript describes immune populations and transcriptomes of individuals with acute SARS-CoV-2 infections and their uninfected contacts, with contrasts among healthy individuals between age groups; between infected and uninfected individuals to describe responses to acute infections; between symptomatic/asymptomatic (per symptom) infected individuals; and correlations drawn between baseline immune responses and convalescent neutralizing antibody titers. Despite its broad scope, the manuscript is clearly written and the data source described includes an age group of great interest (young children <5). These are valuable data and I congratulate the authors on the hard work collecting them.

We thank the reviewer for their supportive comments.

Overarching comments:

1. The largely-naïve cohort is described as a strength, but I'm not sure this is actually a strength of the data. It's unclear how these results inform our understanding of the immune response to SARS-CoV-2 infections in a world with higher population immunity.

The reviewer makes an important point regarding population immunity to SARS-CoV-2. Our analysis is unique in that samples were collected from individuals across the lifespan upon their first interaction with a novel respiratory viral pathogen, and prior to the availability of vaccines. While the immune responses observed in this cohort are likely to differ from those of individuals who have now been vaccinated against the virus or who have a history of prior SARS-CoV-2 infection, they do provide important insights into how age affects virus-induced immune responses among individuals naïve to the virus, including newborn infants. Moreover, immunity varies greatly among individuals depending upon the timing and viral variant of past infections and the timing and product of vaccine doses, making it difficult to disentangle the impacts of age on the immune responses to SARS-CoV-2 infection. Thus, analysis of this cohort enables us to evaluate age as the primary variable modifying immune responses, unclouded by the impact of pre-existing immunity. To more clearly convey these points, we have revised the Introduction and Discussion to describe an important aspect of conducting these analyses on samples from individuals infected early in the COVID-19 pandemic with an ancestral SARS-CoV-2 strain:

Page 5, Lines 96-99: *“SARS-CoV-2-infected participants were enrolled early in the COVID-19 pandemic, prior to the widespread circulation of major variants of concern and the routine availability of COVID-19 vaccines; thus, this analysis focused on immune responses to SARS-CoV-2 among a population that was naïve to the virus.”*

Pages 21-22, Lines 461-466: *“Finally, this analysis focused on participants infected with SARS-CoV-2 early in the pandemic, which may limit the generalizability of our findings to infections caused by currently circulating SARS-CoV-2 variants in a population with substantial herd immunity. Conversely, this SARS-CoV-2-naïve population presents a unique opportunity to understand how individuals respond to SARS-CoV-2 infection in the absence of prior immune experience, and may also provide important insights into how age influences responses to other emerging viral pathogens.”*

2. The lack of a comparator infection, and relatively brief discussion of findings from other infections (only one reference each regarding influenza and RSV) narrows the interpretation of these results. While the paper by Koch et al (reference 20) is limited by its heterogeneous and small cohort, these results (with RSV and influenza infections) merit further discussion/contrast to understand the specificity of the present results to SCV2.

We thank the reviewer for this helpful suggestion. We have revised the Discussion to describe this in more detail:

Pages 19-20, Lines 406-426: *“As we and others have shown, effective mucosal immune responses to respiratory viruses can limit viral replication and reduce systemic immune activation (18, 39). Several prior studies demonstrated a relationship between mucosal immune responses and the severity of respiratory viral infections. In an influenza household transmission study, Oshansky and colleagues reported associations between age, the nasal cytokine response, and disease severity, with innate immune responses in the respiratory mucosa being the strongest predictor of clinical outcomes (40). In a study of 37 children with rhinovirus or respiratory syncytial virus (RSV) lower respiratory infection, García and colleagues found that nasal cytokine concentrations were inversely correlated with disease severity, suggesting a protective effect of a robust immune response within the URT (41). Similarly, Taveras and colleagues observed that children with mild RSV infection had higher levels of mucosal interferons than children with more severe disease (42). Koch and colleagues directly compared host responses of children infected with SARS-CoV-2, influenza, or RSV, and found a high degree of similarity in gene expression within the nasal mucosa across these viral infections (20). RSV and SARS-CoV-2 infections both elicited expression of gene modules associated with inflammatory responses, T cell activation, and IL-6 production, while influenza*

and SARS-CoV-2 infections elicited expression of gene modules associated with antiviral responses, immune cell recruitment, and type I interferon signaling. Notably, upregulation of only one gene module was exclusively associated with SARS-CoV-2 infection; however, this module did not correlate with illness severity or outcomes among SARS-CoV-2-infected children (20). These findings suggest that systemic immune responses may drive differences in the illness severity and outcomes of these viral infections, including across different age groups.”

3. The lack of an upper respiratory specimen from adults is a substantial limitation to the interpretation of age differences in the URT. Each of these contrasts, at most, compares young/school-aged children vs. adolescents, which makes extension of interpretation to broader age-related differences in severity more challenging.

We agree with the reviewer that it was unfortunate that the pediatric and adult nasopharyngeal samples were collected in different media, precluding a direct comparison of host responses in the URT. We have expanded on this limitation in the Discussion section.

Page 21, Lines 454-458: *“Analyses of URT gene expression were limited to children and adolescents because adult nasopharyngeal samples were collected in a medium that did not sufficiently preserve RNA. This lack of data from adult URT precluded evaluation of the relationship between mucosal and systemic immune responses to SARS-CoV-2 in this age group.”*

Specific comments:

1. Counts appear to be off by 1. The authors, in multiple places, refer to a cohort of 201 individuals (137 infected/64 uninfected), but the results ("characteristics of the study population" section) refer to samples collected from 160 children and adolescents recruited through the BRAVE study (111 infected/49 uninfected) and an additional 42 non-hospitalized adults (26 infected/16 uninfected). In the "results" section, that adds up to 202 individuals (137 infected/65 uninfected). Double check.

We thank the reviewer for pointing out this discrepancy. We have reviewed all numbers presented in the manuscript to ensure they are accurate. Analyses were of data from 202 study participants, including 137 individuals with SARS-CoV-2 infection and 65 SARS-CoV-2-uninfected individuals.

2. Unclear in table 1 how many of these infections were medically attended or were infections detected among close contacts from research testing. It's also especially unclear if/how this varies by age. More adults were symptomatic; were they also more likely to be medically attended?

We thank the reviewer for bringing up this point. Healthcare accessibility and utilization were severely disrupted during the first year of the pandemic, and the reasons that individuals who did not have moderate/severe COVID-19 were tested for SARS-CoV-2 or presented for medical evaluation in-person were highly variable and did not correlate well with disease severity. Many patients in this cohort were seen via a telehealth visit to determine their eligibility for SARS-CoV-2 testing; however, this option was largely available only to subjects who had an established medical home at the beginning of the pandemic. Other individuals without an established medical home presented to urgent care clinics or emergency departments seeking testing, regardless of symptom severity, as this was the only way for them to gain access to a healthcare professional. The lack of an established medical home is strongly associated with socioeconomic factors, and many of the groups most severely impacted in the early part of the pandemic were those of lower socioeconomic status, the uninsured, and Hispanic populations, which faced the additional challenge of language barriers for both information about testing and a lack of Spanish-speaking medical personnel. In addition to access challenges, there were constantly evolving guidelines on when individuals should be tested for SARS-CoV-2 and/or who should be evaluated in-person early in the pandemic. For example, very young children and those with chronic respiratory conditions, such as asthma, were more likely to be evaluated in-person due to the perception that they were at greater risk of poor outcomes. Other individuals were required to undergo testing after potential exposures due to their status as essential workers. For these

reasons, we believe data on where SARS-CoV-2 testing was performed and whether individuals were evaluated in-person are unlikely to be a useful measure of disease severity for analyses of this study cohort.

Page 21, Lines 458-461: *“Illness severity was classified based upon symptom prevalence, as testing location and medically attended visits were not reliable measures of disease severity in this cohort of individuals with mild or asymptomatic infections recruited early in the COVID-19 pandemic.”*

3. Unclear if symptom presence/absence was based on any of the timepoints assessed or only the acute timepoint.

We thank the reviewer for bringing up this important point. The presence/absence of symptoms was determined based on follow-up phone calls for up to 28 days after enrollment.

Page 23, Lines 492-494: *“Questionnaires were also administered to assess the presence and duration of specific symptoms at enrollment and at 7, 14, and 28 days after enrollment or until participants reported resolution of all symptoms.”*

4. Among uninfected participants, unclear the timing from exposure (or symptom onset in the infected household member) to specimen collection (other than "max 14 days"). This is especially important for understanding if the "uninfected" groups were just missed infections, since it seems like infections among contacts were determined based on a single swab taken up to 14 days after the index test. This would impact my interpretation of Figures 2 and 3, particularly in the youngest age group with only 10 "healthy" individuals.

The reviewer is correct that we did not re-test study participants after the initial sample collection. However, we did re-contact participants for up to 28 days after enrollment and specifically asked about the development of new symptoms, healthcare encounters, and SARS-CoV-2 testing. Thus, it is likely that the majority of infections would have been identified during these follow-up calls and in subsequent reviews of electronic health records data. To clarify this important point, we have revised the following sentence in the Methods section:

Page 24, Lines 527-530: *“Participants who tested negative for SARS-CoV-2 but who reported one or more symptoms at enrollment or during study follow-up were also excluded because these symptoms could be indicative of false-negative SARS-CoV-2 PCR testing or infection by other adventitious agents.”*

5. Review captions and labelling for sub-panels b, c, and d in Figures 1 and 3.

We have reviewed and verified all captions and labeling in these Figures.

6. In the results (Fig 1A and text): "We observed no significant differences in the proportions of immune cells within the URT of healthy children and adolescents by age (Fig. 1a)." These differences were only observed in the population-adjusted expression analyses, in both the differential gene expression in Fig S1A and in the "modules" described in Fig 1c. On a statistical level, it's unclear if these differences would be significant without accounting for the proportions of immune cells (which are better supported in peripheral blood analyses). On an interpretation level, I'm not clear how this nuance relates to the language in the final sentence of results paragraph 3 ("greater activation of immune cell populations in the URT, which may prime children for..." or the language in the discussion about greater "URT immune tone").

All analyses of upper respiratory gene expression are now presented unadjusted for imputed immune cell populations. Without this adjustment for cell populations, we observe larger (and statistically significant) differences in URT gene expression among healthy children and adolescents by age group (Figure 1), with younger age groups generally having higher expression of innate and adaptive immune modules within the upper respiratory tract. We have revised the Discussion accordingly to reflect the results of these analyses:

Pages 18-19, Lines 383-404: *“The URT is the primary point of entry for SARS-CoV-2 and most other respiratory viruses (37); thus, an improved understanding of host responses to the virus within this niche could identify biological processes that influence infection susceptibility, illness severity, and systemic immune responses to the virus. To date, the majority of studies evaluating host responses to SARS-CoV-2 infection have focused on systemic responses or responses within specific immune cell subsets (38). Analyzing URT samples from healthy children and adolescents without SARS-CoV-2 infection, we found that younger age was associated with broad upregulation of innate and adaptive immune pathways. Only a few prior studies have investigated age-associated differences in mucosal immunity among healthy individuals. Pierce and colleagues performed bulk RNA sequencing of nasopharyngeal samples from 12 children and 27 adults with SARS-CoV-2 infection, observing higher expression of interferon signaling, inflammasome, and innate immune pathways among children relative to adults (19). Of note, the adults in this cohort were more likely to be hospitalized than the children; thus, differences in disease severity or the timing of sampling across age groups could have contributed to the observed differences in URT immune responses (19). Yoshida and colleagues reported that adults with SARS-CoV-2 infection had stronger induction of interferon-related genes within the URT than children; however, this cohort also included patients with varied COVID-19 disease severity (16). Using an age-matched cohort of non-hospitalized, SARS-CoV-2-infected and uninfected children and adults, Loske and colleagues observed that children had greater upregulation of interferon responses within the URT than adults (18). Taken together, these findings suggest that children may be primed to rapidly respond to exogenous pathogens, with the potential for earlier control of infection and reduced illness severity.”*

7. Figure 4/the analysis of symptoms would especially benefit from nuanced engagement with the "medically attended vs. contact" distinction.

We thank the reviewer for bringing up this point. Healthcare accessibility and utilization were severely disrupted during the first year of the pandemic, and the reasons that individuals who did not have moderate/severe COVID-19 were tested for SARS-CoV-2 or presented for medical evaluation in-person were highly variable and did not correlate well with disease severity. Many patients in this cohort were seen via a telehealth visit to determine their eligibility for SARS-CoV-2 testing; however, this option was largely available only to subjects who had an established medical home at the beginning of the pandemic. Other individuals without an established medical home presented to urgent care clinics or emergency departments seeking testing, regardless of symptom severity, as this was the only way for them to gain access to a healthcare professional. The lack of an established medical home is strongly associated with socioeconomic factors, and many of the groups most severely impacted in the early part of the pandemic were those of lower socioeconomic status, the uninsured, and Hispanic populations, which faced the additional challenge of language barriers for both information about testing and a lack of Spanish-speaking medical personnel. In addition to access challenges, there were constantly evolving guidelines on when individuals should be tested for SARS-CoV-2 and/or who should be evaluated in-person early in the pandemic. For example, very young children and those with chronic respiratory conditions, such as asthma, were more likely to be evaluated in-person due to the perception that they were at greater risk of poor outcomes. Other individuals were required to undergo testing after potential exposures due to their status as essential workers. For these reasons, we believe data on where SARS-CoV-2 testing was performed and whether individuals were evaluated in-person are unlikely to be a useful measure of disease severity for analyses of this study cohort.

Page 21, Lines 458-461: *“Illness severity was classified based upon symptom prevalence, as testing location and medically attended visits were not reliable measures of disease severity in this cohort of individuals with mild or asymptomatic infections recruited early in the COVID-19 pandemic.”*

8. In figure 6, there are two observations with high convalescent serum neutralizing activity, who appear to be influencing at least the strength of the associations in this pediatric sample. Are these individuals otherwise outliers in e.g. time to specimen collection (with an IQR of 52-87 days, not sure the range here)?

We thank the reviewer for bringing up this possibility. Interestingly, the two individuals identified by the reviewer were not outliers in terms of their demographics, illness characteristics, or timing of sample collection. However, we determined that the observed correlations between acute URT responses and convalescent serum neutralizing activity were no longer statistically significant after exclusion of the data from these two outliers. Based on this, we believe it is prudent to remove this analysis from the current manuscript. We plan to revisit these analyses when additional data on long-term humoral immune responses are available from these cohorts.

Response to Reviewers

Reviewer #1 (Remarks to the Author):

I am satisfied with the revisions that were made on the manuscript.

Editorial note: This reviewer was additionally asked to comment in the place of reviewer 2 and 3 who were unable to provide a response in this round of review.

I've had a look at this and my view is the authors have adequately addressed the reviewers' comments.

We thank the reviewer for their time evaluating this article and for their constructive comments, which have significantly strengthened the manuscript.

Reviewer #4 (Remarks to the Author):

Regarding the response to reviewer #2, the authors sufficiently highlighted the aspects analyzed in this study, particularly the opportunity to understand innate and adaptive immune pathways in the UTR in a SARS-CoV-2 naive pediatric and adult population. This is a large cohort which targets the early phase of the pandemic, although not generalizable to the situation we have now with different history of vaccination and contact to circulating VOCs such as omicron, however, the study constitutes an important contribution to understand viral infection by SARS-CoV-2 in general and to allow future analogous deductions from this findings in the case new viral threats emerge.

We thank the reviewer for their time evaluating the manuscript and for their supportive comments.

Additions by table S1 are helpful to understand the cohort analyzed in the study. All of the major comments regarding illustration of the data and showing data on both pediatric and adult groups as well as giving details about the methods used and handling of samples have been addressed in the revised manuscript.

We agree with the Reviewer that the additional details in Table S1 and additional details in the methods have enhanced the presentation of the study data.

Of course, single cell sequencing and other modern methods may have elucidated more details on the single cell pathways, however, considering the important contribution by analyzing a pediatric and adult SARS-CoV-2-naive populations at their first contacts to SARS-CoV-2 is worth to be published as analogical conclusions may be derived from this study in the case new viral pathogens emerge.

We agree with the Reviewer that single cell sequencing would have provided additional detail, but agree that analyzing a naïve study population can provide new insights into the response to novel viral pathogens.

Regarding reviewer #3 I agree that the cohort is rather small to completely understand the anti-viral mechanisms in children and adults. However, the data seem worth to be published as they represent a real-world situation with high inter-individual variance. The concerns of reviewer #3 have been accordingly addressed by the authors and included into the discussion section. Limitations of the study particularly regarding the methods and the generalizability of results to the current situation are sufficiently discussed in the revised manuscript.

We thank the reviewer for their supportive comments.